# AIRUSE-LIFE +: Estimation of natural source contributions to urban ambient air PM₁₀ and PM₂.₅ concentrations in Southern Europe. Implications to compliance with limit values.

[5] Evangelia Diapouli[1], Manousos I. Manousakas[1], Stergios Vratolis[1], Vasiliki Vasilatou[1], Stella Pateraki[1], Kyriaki A. Bairachtari[1], Xavier Querol[2], Fulvio Amato[2], Andrés Alastuey[2], Angeliki A. Karanasiou[2], Franco Lucarelli[3], Silvia Nava[3], Giulia Calzolai[3], Vorne L. Gianelle[4], Cristina Colombi[4], Célia Alves[5], Danilo Custódio[5], Casimiro Pio[5], Christos Spyrou[6], George B. Kallos[6], Konstantinos Eleftheriadis[1]

[1] Institute of Nuclear & Radiological Science & Technology, Energy & Safety, N.C.S.R. "Demokritos", Athens, 15341, Greece
[10] [2] Institute of Environmental Assessment and Water Research (IDAEA-CSIC), Barcelona, 08034, Spain
[3]Department of Physics and Astronomy, Università di Firenze and INFN-Firenze, Sesto Fiorentino, 50019, Italy
[4]Environmental Monitoring Sector, Arpa Lombardia, Milano, I-20124, Italy
[5]Centre for Environmental & Marine Studies, Department of Environment, University of Aveiro, Aveiro, 3810-193, Portugal
[6]School of Physics, University of Athens, Athens, 15784, Greece

[15] *Correspondence to*: Evangelia Diapouli (ldiapouli@ipta.demokritos.gr)

## Abstract

The contribution of natural sources to ambient air particulate matter (PM) concentrations is often not considered; however, it may be significant for certain areas and during specific periods of the year. In the framework of the AIRUSE-LIFE+ project, state-of-the-art methods have been employed for assessing the contribution of major natural sources (African dust, sea salt and

[20] forest fires) to PM concentrations, in Southern European urban areas. 24 h measurements of PM₁₀ and PM₂.₅ mass and chemical composition were performed over the course of a year in five cities: Porto, Barcelona, Milan, Florence and Athens. Net African dust and sea-salt concentrations were calculated based on the methodologies proposed by EC (SEC 2011/208). The contribution of uncontrolled forest fires was calculated through receptor modelling. Sensitivity analysis with respect to the calculation of African dust was also performed, in order to identify major parameters affecting the estimated net dust

[25] concentrations. African dust contribution to PM concentrations was more pronounced in the Eastern Mediterranean, with the mean annual relative contribution to PM₁₀ decreasing from 21 % in Athens, to 5 % in Florence, and around 2 % in Milan, Barcelona and Porto. The respective contribution to PM₂.₅ was calculated equal to 14 % in Athens and from 1.3 to 2.4 % in all other cities. High seasonal variability of contributions was observed, with dust transport events occurring at different periods in the Western and Eastern Mediterranean basin. Sea salt was mostly related to the coarse mode and also exhibited significant

[30] seasonal variability. Sea-salt concentrations were highest in Porto, with average relative contributions equal to 12.3 % for PM₁₀. Contributions from uncontrolled forest fires were quantified only for Porto and were low on an annual basis (1.4 % and

1.9 % to PM$_{10}$ and PM$_{2.5}$, respectively); nevertheless, contributions were greatly increased during events, reaching 20 and 22 % of 24 h PM$_{10}$ and PM$_{2.5}$ concentrations, respectively.

## 1 Introduction

Atmospheric aerosols may be emitted by both natural and anthropogenic sources. Given that exposure to particulate matter (PM) is mainly related to urban environments where anthropogenic activities lead to increased concentration levels, natural sources are often not considered. Nevertheless, their contribution may be significant, especially for certain areas and during specific periods of the year. It has been estimated that the natural contribution to PM may range from 5 % to 50 % in different European countries (Marelli, 2007). Background annual average PM$_{10}$ mass concentration for continental Europe is 7.0 ± 4.1 µg m$^{-3}$ (Van Dingenen et al., 2004) and is attributed both to natural sources and anthropogenic long-range transported particles. This background level shows regional variations, and in some cases (in particular for the Southern European countries) naturally emitted PM may contribute significantly, causing even exceedances of air quality standards (Pey et al., 2013; Querol et al., 2009; Rodriguez et al., 2001; Querol et al., 1998). The main natural sources affecting ambient PM levels are wind-blown soil dust, sea salt, wildfires, volcanic ash and biogenic aerosol (Viana et al., 2014).

Wind-blown soil dust relates to the transport of mineral dust particles from agricultural and arid or semiarid regions (Ginoux et al., 2012). North Africa is the main source of desert dust for European countries (Stuut et al., 2009). Most of these particles are very coarse (diameter ≥10 µm) and are thus deposited close to the source region, while a significant amount of coarse particles (diameter around 1-10 µm) can be transported over long distances. An estimation of the emission flux of desert aerosols that is subject to long-range transport is of the order of 1500 Tg/yr (Papayannis et al., 2005).

Sea-salt aerosol is emitted from the sea surface, through bubble-bursting processes resulting in sea-spray particles with sizes ranging from sub micrometre to a few micrometres (O'Dowd and de Leeuw, 2007). Sea-salt aerosols play an important role in atmospheric chemistry, providing the surface for heterogeneous reactions and acting as a sink for anthropogenic and natural gaseous pollutants (Tsyro et al., 2011). The presence of sea-salt aerosols in the atmosphere was shown to significantly alter the regional distribution of other inorganic aerosols, namely sulphate, nitrate and ammonium (van den Berg et al., 2000). It may also appear in both the coarse and fine fraction (Eleftheriadis et al., 2014). Furthermore, sea salt helps to reduce the acidity of the air by providing base cations (Tsyro et al., 2011).

Wildfires relate to the burning of forests and other vegetation, mostly through natural processes. Large-scale forest fires are a major PM source, while smoke plumes may be transported over thousands of kilometres, affecting air quality at local, regional and global scale (Faustini et al., 2015; Diapouli et al., 2014). Volcanic ash can also have a global impact due to the fact that emissions may be injected into the stratosphere but have more infrequent occurrence (von Glasow et al., 2009). Biogenic aerosol is emitted by vegetation and may be of primary or secondary origin (Caseiro et al., 2007).

Taking into account that natural sources cannot be controlled, while their contribution varies between the European countries, EU legislation has allowed for the subtraction of PM concentrations of natural origin when Member-States assess and report attainment of air quality standards. Apart from environmental reporting, quantification of natural contributions to PM levels is important in terms of exposure assessment as well. Many epidemiological studies have demonstrated the detrimental effects of particulate matter pollutants to human health (Ostro et al., 2015; Samoli et al., 2013). The distinct physico-chemical and toxic properties of anthropogenic and naturally emitted aerosol call for a differentiation of peak concentration days due to anthropogenic pollution or natural events, when assessing population exposure and dose-effect relationships. On the other hand, extreme natural events that lead to very high exposures may still adversely affect human health, especially in the case of exposures on markedly different aerosol size fractions (Zwozdziak et al., 2017) or sensitive population subgroups (Perez et al., 2008).

High background concentration levels are frequently reported in Southern European countries, often due to the enhanced contribution by natural sources. The Mediterranean climate, characterized by increased solar radiation and low rainfall rates, promotes aerosol production and reduces the potential for dispersion and removal of pollutants (Lazaridis et al., 2005). The vicinity of Southern Europe to North Africa on the other hand results in frequent and intense dust outbreaks, with high loads of dust from desert regions transported across the Mediterranean, which often leads to exceedances of air quality limit values (Pey et al., 2013; Nava et al., 2012; Athanasopoulou et al., 2010; Querol et al., 2009; Gerasopoulos et al., 2006; Kallos et al., 2006; Rodriguez et al., 2001; Querol et al., 1998).

In the framework of the AIRUSE-LIFE+ project, the contribution of major natural sources to $PM_{10}$ and $PM_{2.5}$ concentration levels was quantified for five Southern European cities (in Portugal, Spain, Italy and Greece). The project focused on two sources: the long-range transport of African dust and sea salt. The contribution from wildfires has been also detected and quantified in one city (Porto). In addition, a sensitivity analysis on the calculation of African dust contributions was performed, providing useful insight into the key factors affecting the quantified dust concentrations.

## 2 Experimental methods

### 2.1 Sampling and analysis

Year-long measurement campaigns were performed from January 2013 to February 2014 in five Southern European cities: Porto, Barcelona, Milan, Florence and Athens (Fig. 1). The cities were selected in order to cover Southern Europe from West to East, as well as sites by the sea and inland. 24 h sampling of $PM_{10}$ and $PM_{2.5}$ (00:00 – 23:59) was performed once every 3 days for a full year in all cities. Additional sampling was conducted during days when African dust episodes were forecasted, in order to better characterize the contribution of this source to PM levels. Comprehensive chemical characterization of $PM_{10}$

and $PM_{2.5}$ samples was performed for the determination of organic and elemental carbon, carbonate carbon, levoglucosan, ion species and major and trace elements. The measurement periods, monitoring sites and number of valid chemical speciation samples for each city are presented in Table 1. Details about sites, sampling and analytical procedures are provided in detail in Amato et al. (2016).

## 2.2 Quantification of the contribution of natural sources

Contributions of sea salt and African dust to $PM_{10}$ and $PM_{2.5}$ concentrations were quantified based on EU guidelines (SEC 2011/208) for all five AIRUSE cities. Specifically for African dust, potential African dust transport events at each city were identified through: (i) 5-day backward air mass trajectories obtained every 3 hours and at 3 heights (500, 1000 and 1500 m a.s.l.) by the Hybrid Single Particle Lagrangian Integrated Trajectory (HYSPLIT) Model (Draxler and Rolph, 2003); (ii) dust load surface concentrations provided by the Barcelona Supercomputing Centre (BSC)-DREAM8b v2.0 Atmospheric Dust Forecast System (Basart et al., 2012); (iii) dust concentrations at surface levels and at three additional heights (reaching up to ~950 m a.s.l.) provided by SKIRON/Dust forecast model (Spyrou et al., 2010); (iv) 7-day backward air mass trajectories obtained every 6 hours and at 3 heights (500, 1000 and 1500 m a.s.l.) by the Flextra model (Stohl and Seibert, 1998). Following the identification of days potentially affected by long-range transport of African dust, net African dust concentrations were calculated based on continuous 24 h PM data from background sites representative of the regional background concentrations at the studied cities. $PM_{10}$ and $PM_{2.5}$ data available from the National Monitoring Networks operating at the five cities were used for this analysis. 30-days moving averages of the previous and next 15 days of the regional background concentrations were calculated, excluding days with potential African dust transport. Averages corresponded to 40[th] percentiles in the case of Porto, Barcelona and Florence (Escudero et al., 2007). In Milan and Athens, a more conservative indicator, the 50th percentile, was selected since it was found to reproduce better PM background concentrations (SEC 2011/208). Net African dust load was quantified for each day forecasted as potential dust event by at least one of the above mentioned models, as the observed increase in concentration with respect to the calculated moving average for that day (representative of background concentration not affected by dust transport).

Sea salt (ss) was calculated based on major sea-salt components (Cl and Na) and typical elemental ratios for sea water (Mg/Na, K/Na, Ca/Na and $SO_4^{2-}$/Na) and the earth's crust (Na/Al) (Calzolai et al., 2015):

$$Sea\ salt = [ssNa] + [Cl] + [ssMg] + [ssK] + [ssCa] + [ssSO_4^{2-}], \qquad (1)$$

where:

$[ssNa] = [Na] - [nssNa]$

$[nssNa] = 0.348{\times}[Al]$

$[ssMg] = 0.119{\times}[ssNa]$

$$[ssK] = 0.037 \times [ssNa]$$
$$[ssCa] = 0.038 \times [ssNa]$$
$$[ssSO_4{}^{2-}] = 0.253 \times [ssNa].$$

The contribution of wildfires was estimated only for Porto, where several wildfires were registered during late August and September of 2013. A biomass burning factor was obtained by receptor modelling (Positive Matrix Factorization, PMF), with several peak concentrations during the wildfires' period; thus, these concentrations were attributed to wildfires and classified as natural source contributions. Details on the PMF analysis and results are presented in Amato et al. (2016).

**2.3 Sensitivity analysis on the estimation of African dust contribution**

A sensitivity analysis was performed in order to assess the main parameters affecting the quantification of net African dust concentrations. Specifically, the following parameters were examined with respect to the calculation of net African dust: (i) the identification of dust transport episodes by different modelling tools; (ii) the use of PMF analysis for the identification of a mixed mineral dust source or a separate African dust source; (iii) the use of alternative input concentration data, such as the coarse PM fraction ($PM_{2.5-10}$) and the mineral component of $PM_{10}$, calculated either by PMF analysis or reconstructed from elemental concentrations based on stoichiometry (Nava et al., 2012; Marcazzan et al., 2001):

$$[Mineral\ dust] = 1.15 \times (1.89 \times [Al] + 2.14 \times [Si] + 1.67 \times [Ti] + 1.4 \times [soilCa] + 1.2 \times [soilK] + 1.4 \times [soilFe])$$

$$(2)$$

where the soil fractions of Ca, K and Fe have been calculated using their typical crustal ratios with respect to Al (Mason, 1966): [soilCa]=0.45×[Al], [soilK]=0.32×[Al], [soilFe]=0.62×[Al].

Net dust concentrations calculated by $PM_{10}$ regional background data were also compared to the dust concentrations provided by the SKIRON/Dust and BSC - DREAM8b v2.0 transport models.

**3 Results and discussion**

**3.1 Contribution of natural sources to PM concentrations and exceedances**

Mean annual contributions of long-range transported African dust, sea salt and wildfires (estimated only for Porto) to $PM_{10}$ and $PM_{2.5}$ concentrations at each city are presented in Tables 2 and 3, along with their respective uncertainties. African dust and sea-salt concentrations were calculated based on SEC 2011/208. Only in the case of Florence, where PMF analysis produced a separate source attributed to the long-range transport of Saharan dust, the concentrations of African dust for $PM_{10}$ and $PM_{2.5}$ reported in Tables 2 and 3 correspond to the contributions of the Saharan dust PMF factor. The contribution of

wildfires in Porto was also estimated by PMF analysis. The uncertainties of the African dust and sea-salt concentrations were calculated based on the uncertainties of the parameters included in the respective calculation formulas (PM regional concentrations in the case of African dust and Na, Cl and Al concentrations for sea salt). The uncertainties associated with PMF analysis (contribution of African dust in Florence and of wildfires in Porto) were calculated based on the standard error of the coefficients of a multiple regression between the measured PM concentration (independent variable) and the source contributions estimated by PMF analysis (dependent variables).

The African dust contribution to PM concentrations was found to be more pronounced in the Eastern Mediterranean (Athens), with peak concentrations during spring time reaching up to 127 µg m$^{-3}$ (maximum 24 h mean dust concentration during a 15-day dust transport event in May 2013). Previous studies have also reported the high impact of dust transport events in Athens and Greece in general (Manousakas et al., 2015; Grigoropoulos et al., 2009; Mitsakou et al., 2008). The mean annual relative contributions of African dust to the PM$_{10}$ concentrations decreased from East to West: 21 % in Athens, 5 % in Florence, and ~ 2 % in Milan, Barcelona and Porto. The respective contributions to the PM$_{2.5}$ concentrations were 13.7 % in Athens, 1.3-1.4 % in Florence and Milan and 2.3-2.4 % in Barcelona and Porto. The large difference between the net dust loads calculated for Athens and the other cities is due to the Southern location of Athens, and the severity of some Saharan dust episodes in the eastern part of the Basin (Athanasopoulou et al., 2016). A high seasonal variability of contributions was observed, with dust transport events occurring at different periods in the western and eastern sides of the Mediterranean (Pey et al., 2013; Querol et al., 2009). The African dust inputs were highest during spring and lowest during summer in Athens and Florence. Milan presented high contributions during spring and summer, Porto during winter and Barcelona during the summer season. These results are in good agreement with Moulin et al. (1998) who reported that the annual cycle of African dust transport over the Mediterranean region starts during springtime in the eastern part, while during summer there is maximum transport in the western part. Porto was the only city deviating from this behaviour, suggesting that the studied year may not be representative for this city for assessing seasonal trends, probably due to the low frequency and intensity of dust events. Querol et al. (2009) have also noted that when intense dust transport events are recorded in the Eastern Mediterranean (such as the case for 2013), unusually low African dust contributions are observed in the Western Mediterranean. Sea salt was mostly related to the coarse mode and exhibited significant seasonal variability as well. The sea-salt concentrations were highest in Porto, with average relative contributions equal to 12.3 % and 4.6 % for PM$_{10}$ and PM$_{2.5}$. The respective contributions for Athens and Barcelona were 7–8 % to PM$_{10}$ and 2.3-2.5 % to PM$_{2.5}$. The lowest contributions were observed in Florence and Milan (1.3-3.3 % to PM$_{10}$). The results reflect the geographical distribution of the AIRUSE sites: lower levels of sea salt at the inland Italian cities (Florence and Milan) and higher at the Mediterranean coastal sites, with the highest contribution observed at the Atlantic site (Porto). Similar observations were reported by Manders et al. (2010) who showed that the sea-salt load in PM$_{10}$ at the Atlantic side of Europe is much higher than in the Mediterranean region, especially the western Mediterranean. They also showed that the sea-salt load in PM$_{10}$ is reduced very fast as the air masses progress inland.

Large-scale uncontrolled forest fires were observed only in Porto during the period of the study. The average contribution to the PM levels was low (1.4 % and 1.9 % to $PM_{10}$ and $PM_{2.5}$, respectively) due to the few event days during the year (after the 20th of August and during September). Nevertheless, during event days, the contribution to PM was greatly increased, reaching 20 and 22 % to $PM_{10}$ and $PM_{2.5}$, respectively.

The uncertainties for the calculated contributions of the different natural sources were estimated on average around or below 10%. The relative uncertainties exhibited low variability during the studied period, except for the case of African dust, where a significant increase was observed for net dust concentrations below 5 µg m$^{-3}$. The relative uncertainties calculated for each city and PM size fraction were on average at 6-15%, 10-41% and above 100% for African dust loads above 5 µg m$^{-3}$, between 1 and 5 µg m$^{-3}$ and below 1 µg m$^{-3}$, respectively.

The subtraction of the contribution of natural sources from the $PM_{10}$ concentrations measured at the AIRUSE sites, according to EC regulation, led to a decrease in the mean annual $PM_{10}$ concentrations in the range of 3.5 (Milan) to 29.5 % (Athens) (Fig. 2). Attainment of the annual limit value set by the EU through Directive 2008/50/EC was achieved at all sites during 2013, although the urban background site in Milan and the urban traffic site in Porto exhibited concentrations close to the air quality standard. A similar decrease (1.5–21 %) was observed in the 90.4th percentiles of $PM_{10}$ concentrations. The 90.4th percentile
corresponds to the maximum permissible number of exceedance days (35 during the year). The subtraction of the contribution of natural sources led to a marginal compliance with the 24 h limit value for Porto, while Milan continued to present more exceedances than the permitted 35 days (84 days for $PM_{10}$ and 82 days for the adjusted $PM_{10}$ after subtraction of the contribution of natural sources). Regarding $PM_{2.5}$ concentrations, the subtraction of the contribution of natural sources led to decreases in mean annual concentrations in the range of 1.3 (Florence) to 16 % (Athens) for the AIRUSE sites. Despite the
subtraction of the contribution of natural sources, Milan did not attain the EU annual limit value, while in the urban traffic site in Porto, marginal attainment was achieved (Fig. 3).

The average contributions of natural sources to the $PM_{10}$ concentrations at each city, during all measurement days and only when exceedance days were considered, are presented in Fig. 4. Wildfires contributed to exceedances in Porto. The average concentration during exceedance days was low (below 4 µg m$^{-3}$), nevertheless it was much higher than the corresponding mean
value during the yearly measurement campaign. Sea salt, on the other hand, is related to clean air conditions, while no African dust event was recorded during exceedance days. In the Barcelona urban background site, 24 h concentrations did not exceed the respective EU limit. The highest concentrations were almost entirely attributed to anthropogenic sources. Again no dust events were recorded during high concentration days. Similar results were obtained for Florence and Milan as well. The Athens suburban site on the other hand is a characteristic example of the effect of natural sources in background urban environments.
The exceedances of the $PM_{10}$ 24 h EU limit value were attributed to African dust by 79% in terms of mass concentration (53 out of 67 µg m$^{-3}$), with a total contribution from natural sources reaching 88%. The mean annual contribution of African dust was also significant (21%).

## 3.2 Sensitivity analysis on the estimation of net African dust

Based on the available tools for dust transport modelling, different potential dust event days may be identified. Analysis of the AIRUSE data showed that the results from the various models are not always in perfect agreement. A sensitivity analysis was performed in order to assess the effect of model selection, based on the Athens dataset, which included the largest number of dust events. Net African dust loads were calculated using SEC 2011/208. In this analysis, days were marked as dust events for the following scenarios: (N1) when at least 1 out of 4 models gave an event signal; (N2) when at least 2 models gave an event signal; (N3) when at least 3 models gave an event signal; (N4) when all models gave an event signal. The results of the calculated dust concentrations for each of the scenarios (N2) - (N4) (shown in blue) and the respective increments (shown in red) when a less strict criterion is selected, (N1) – (N3) respectively, are presented in Fig. 5.

Small increments in relation to peak dust concentrations were observed between (N1), (N2) and (N3) scenarios, with mean annual dust contribution calculated equal to 5.1, 4.3 and 3.9 µg m$^{-3}$ for scenarios (N1)–(N3) respectively. Nevertheless, on a daily basis, these increments reached up to 16 µg m$^{-3}$ for scenario (N1) in relation to (N2) and 25 µg m$^{-3}$ for scenario (N2) in relation to (N3), which are of the same magnitude of typical PM$_{10}$ concentration levels at this site (Triantafyllou et al., 2016). When full agreement between models was required, even very intense events were omitted, as is demonstrated by the comparison of the (N3) and (N4) scenarios. The analysis highlights the need for employing as many available tools as possible for the identification of dust transport events, in order to ensure adequate coverage and reduce uncertainty.

Another parameter examined was the use of alternative input data in the net dust calculation algorithm. The net dust loads calculated by PM$_{10}$ regional background concentrations according to the methodology adopted by EC, Net dust (PM$_{10}$), were used as reference. Net dust loads were also calculated by using the following input datasets: i) the coarse fraction of PM (PM$_{2.5-10}$) regional concentrations (instead of the PM$_{10}$ fraction), Net dust (PMcoarse), and the mineral component of PM$_{10}$, ii) either reconstructed through stoichiometry, Net dust (MIN-STOICH), or iii) obtained by PMF, Net dust (PMF).

PMF analysis performed on the datasets of all five studied cities, reported in Amato et al. (2016), has shown that a distinct African dust factor is not easily obtained. Only in the case of Florence, a separate PMF factor profile for African dust was identified, providing a potential reference value for this city and insight into the chemical profile of transported African dust. The African dust concentrations estimated by PMF in Florence are in very good agreement with the Net dust (MIN-STOICH), while the method based on PM$_{10}$ concentrations at the regional site seem to overestimate African dust loads (Fig. 6). This last observation may be related to the difficulty of finding a suitable regional background site representative for the city of Florence in connection to African dust transport, due to the orography of the region.

In all other AIRUSE cities, a mixed mineral dust factor was obtained, including both local soil and long-range transported dust. Comparisons of the net dust loads calculated based on the mineral component of PM$_{10}$ (quantified stoichiometrically or by PMF) with the reference Net dust (PM$_{10}$), for the city of Athens, are shown in Fig. 7. For Porto, Barcelona and Milan no

regression between Net dust (MIN-STOICH) or Net dust (PMF) with Net dust ($PM_{10}$) was attempted, due to the much lower number of African dust event days and corresponding chemical speciation data. For the ATH-SUB dataset, the use of the mineral dust contributions estimated by PMF provided results in good agreement with Net dust ($PM_{10}$) concentrations, with the uncertainty increasing in dust concentrations below 10 µg $m^{-3}$. The net dust calculated from the $PM_{10}$ stoichiometric mineral

component (MIN-STOICH) exhibited very good correlation with Net dust ($PM_{10}$). Net dust (MIN-STOICH) displayed lower concentrations by a factor of 1.6 on average, while for net dust loads <10 µg $m^{-3}$ this difference was higher (Fig. 7). Similar behaviour, with an even higher correlation coefficient, was observed when PMcoarse concentrations were used in the calculation algorithm (Net dust (PMcoarse)) (Fig. 8). Barcelona exhibited comparable results with Athens (Fig. 8), while weaker correlations were observed for Porto and Milan (Fig. 9). Florence was not included in this analysis because no $PM_{2.5}$

or PMcoarse data were available from the regional background site of the National Monitoring Network. The results indicate that African dust is also found in sizes below 2.5 µm.

Regression analysis of the calculated Net dust ($PM_{10}$) and net dust (MIN-STOICH) versus $PM_{2.5}$/$PM_{10}$ concentration ratios was used in order to further examine the calculated dust loads with respect to particle size (Fig. 10). In the case of Net dust (MIN-STOICH), all intense dust events (with net dust loads greater than 10 µg $m^{-3}$) were related with the coarse fraction (low

$PM_{2.5}$/$PM_{10}$ ratios). On the contrary, for Net dust ($PM_{10}$) several events with net dust loads from 10 to 20 µg $m^{-3}$ or higher were related to fine particles ($PM_{2.5}$/$PM_{10}$ ratios greater than 0.6). This suggests that Net dust ($PM_{10}$) may include non-mineral fine particles.

The chemical profiles of mineral dust obtained by PMF (Amato et al., 2016) may provide further information on the discrepancies observed between the alternative methods. Comparison of the Athens mineral dust profile and the two mineral

dust profiles obtained for Florence (for local and African dust), showed that the African dust profile differed with respect to the other two mineral dust profiles in the absence of organic carbon, Zn and Pb, while a much lower $NO_3^-$ contribution was also observed. The presence of these species may reflect the enrichment of local dust with anthropogenic chemical components. On the other hand, the inclusion of non-mineral components in the African dust profile (Fig. 11) may explain the underestimation in Athens of net dust loads when the $PM_{10}$ mineral component, Net dust (MIN-STOICH), is used (Rodriguez

et al., 2001).

Net dust concentrations calculated by $PM_{10}$ regional background data were also compared to the dust concentrations provided by the SKIRON/Dust model (at surface level and at three different heights) and the BSC-DREAM8b v2.0 model (at surface level). Very good correlation was obtained with the Athens dataset for both models. For the SKIRON/Dust model, the calculated and modelled dust loads at surface levels were comparable (Table 4). Nevertheless, no correlation was observed for

dust concentrations below 10 µg $m^{-3}$, suggesting increased uncertainty at these dust levels (Fig. 12) possibly due to the applied dust cycle parametrization constraints and limitations. In the case of Porto, Barcelona and Milan, almost all modelled dust concentrations were below or equal to 10 µg $m^{-3}$, thus producing weak to no correlations with calculated dust loads. Florence

presented similar results as Athens, with somewhat lower Pearson's coefficients between modelled and calculated data, which may be attributed to fewer data with concentrations above 10 μg m$^{-3}$. In addition, the corresponding slopes of the regression lines were higher than 1.0 in all cases (Table 4). The differences observed in the slopes and intercepts calculated for SKIRON/Dust and BSC-DREAM8b v2.0 models are related to the parametrizations used by each model for simulating the desert dust cycle, and more specifically with respect to the dust uptake scheme and the soil characterization.

## 4 Conclusions

The LIFE-AIRUSE project employed a large dataset of PM$_{10}$ and PM$_{2.5}$ concentrations and chemical speciation from five Southern European cities (Porto, Barcelona, Milan, Florence and Athens), in order to examine the contribution of two major natural sources: long-range transport of African dust and sea salt. The results clearly show that the natural source contribution may be significant during specific periods, leading to events of PM limit value exceedances. The African dust contribution to the PM concentrations was more pronounced in the Eastern Mediterranean (Athens), with peak 24 h concentrations in spring time reaching up to 127 μg m$^{-3}$ during a 15-day long African dust event in May 2013. The mean annual relative contributions of African dust to the PM$_{10}$ concentrations decreased from East to West: 21 % in Athens, 5 % in Florence, and ~ 2 % in Milan, Barcelona and Porto. High seasonal variability of contributions was observed, with dust transport events occurring at different periods in the western and eastern sides of the Mediterranean. Sea salt was mostly related to the coarse mode and exhibited significant seasonal variability. The sea-salt concentrations were highest in Porto, with average relative contributions equal to 12.3 % for PM$_{10}$. The respective contributions for Athens and Barcelona were 7–8 %, while the lowest contributions were observed in Florence and Milan (1.3-3.3 %). The results reflect the geographical distribution of the AIRUSE sites: lower levels of sea salt at the inland Italian cities (Florence and Milan) and higher at the Mediterranean coastal sites, with the highest contribution observed at the Atlantic site (Porto). Uncontrolled forest fires were observed to affect PM concentrations only in Porto during the studied period. The mean annual contribution to the PM levels was low (1.4 % and 1.9 % to PM$_{10}$ and PM$_{2.5}$, respectively) due to the few event days during the year (after the 20$^{th}$ of August and during September). Nevertheless, during event days, the contribution to PM was greatly increased, reaching 20 and 22 % of 24 h PM$_{10}$ and PM$_{2.5}$, respectively.

A sensitivity analysis for the quantification of African dust contribution was performed, in order to assess the major factors affecting the calculated net dust concentrations. The analysis indicated that a key parameter to be considered is the selection of an appropriate regional background site. In addition, the use of as many available tools as possible for the identification of dust transport events is recommended, in order to ensure adequate coverage and reduce uncertainty. The results also indicated that the calculation of net African dust through the use of regional background data of PM$_{10}$ (or PM$_{2.5}$) mass concentrations provides higher dust concentration estimates in comparison to the use of the same methodology with as input data the mineral component of PM, derived stoichiometrically. The analysis of mineral dust source profiles obtained by PMF provides further

evidence that additional species to the crustal matter, usually secondary aerosol, are the source of this discrepancy, arriving together or associated with the crustal component during long-range transport.

The present study has demonstrated that natural sources are often expressed with high intensity events, leading to very high daily contributions and exceedances of the EU air quality standards. Since these sources cannot be controlled, relevant mitigation measures can only be focused on minimizing the effects of this type of pollution. Namely, measures are recommended to target reducing the potential of particles deposited on the streets and other surfaces to resuspend, while emergency action plans, especially for sensitive population subgroups, may come into force during days when extreme dust events are forecasted.

**Acknowledgments**

The authors gratefully acknowledge: (i) the NOAA Air Resources Laboratory (ARL) for the provision of the HYSPLIT transport and dispersion model and READY website (http://www.ready.noaa.gov); (ii) the Barcelona Supercomputing Centre for the provision of the BSC-DREAM8b (Dust REgional Atmospheric Model) model (http://www.bsc.es/projects/earthscience/BSC-DREAM/); (iii) Andreas Stohl (NILU), Gerhard Wotawa and Petra Seibert (Institute of Meteorology and Geophysics, Vienna) and ECMWF (European Centre for Medium Range Weather Forecast) for the provision of FLEXTRA model.

This work was funded by the AIRUSE LIFE+ (ENV/ES/584) EU project. Financial support from the EnTeC FP7 Capacities programme (REGPOT-2012-2013-1, FP7, ID: 316173) projects is also acknowledged.

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

**Table 1.** Description of measurement campaigns: Measurement sites, periods and sampling days.

| Monitoring site | Site acronym | Measurement period | Number of samples |
|---|---|---|---|
| Porto<br>Urban traffic | POR-TR | 01/2013-01/2014 | 122 ($PM_{10}$) / 125 ($PM_{2.5}$) |
| Barcelona<br>Urban background | BCN-UB | 01/2013-01/2014 | 125 ($PM_{10}$) / 109 ($PM_{2.5}$) |
| Milan<br>Urban background | MLN-UB | 01/2013-01/2014 | 276 ($PM_{10}$) / 357 ($PM_{2.5}$) |
| Florence<br>Urban background | FI-UB | 01/2013-01/2014 | 223 ($PM_{10}$) / 243 ($PM_{2.5}$) |
| Athens<br>Suburban | ATH-SUB | 02/2013-02/2014 | 192 ($PM_{10}$) / 212 ($PM_{2.5}$) |

**Table 2.** Mean annual natural source contributions to the $PM_{10}$ concentrations and corresponding uncertainties, for the five AIRUSE cities.

| | Contributions of natural sources (µg m$^{-3}$)[1] | | | | |
|---|---|---|---|---|---|
| | **Porto** | **Barcelona** | **Milan** | **Florence** | **Athens** |
| $PM_{10}$ concentration | 34.6 | 22.5 | 35.8 | 19.8 | 19.6 |
| African dust | 0.75 ± 0.02 | 0.49 ± 0.01 | 0.76 ± 0.02 | 1.02± 0.14 | 4.19 ± 0.55 |
| Sea salt | 4.27 ± 0.41 | 1.49 ± 0.18 | 0.46 ± 0.03 | 0.64 ± 0.07 | 1.64 ± 0.14 |
| Wildfires | 0.50 ± 0.02 | NE[2] | NE[2] | NE[2] | NE[2] |
| **Total natural** | **5.52 ± 0.45** | **1.98 ± 0.19** | **1.22 ± 0.05** | **1.66 ± 0.21** | **5.83 ± 0.69** |
| | Relative contributions of natural sources (%) | | | | |
| African dust | 2.2 ± 0.1 | 2.2 ± 0.0 | 2.2 ± 0.0 | 5.2 ± 0.7 | 21.4 ± 2.8 |
| Sea salt | 12.3 ± 1,2 | 6.6 ± 0.8 | 1.3 ± 0.1 | 3.3 ± 0.3 | 8.1 ± 0.7 |
| Wildfires | 1.4 ± 0.1 | NE[2] | NE[2] | NE[2] | NE[2] |
| **Total natural** | **15.9 ± 1.4** | **8.8 ± 0.8** | **3.5 ± 0.1** | **8.5 ± 1.0** | **29.5 ± 3.5** |

[1]Contributions may slightly differ from the values reported in Amato et al. (2016) due to different statistics of the respective datasets.

5    [2]NE: Not estimated.

**Table 3.** Mean annual natural source contributions to the PM$_{2.5}$ concentrations and corresponding uncertainties, for the five AIRUSE cities.

| | Contributions of natural sources (µg m$^{-3}$)[1] | | | | |
|---|---|---|---|---|---|
| | **Porto** | **Barcelona** | **Milan** | **Florence** | **Athens** |
| PM$_{2.5}$ concentration | 26.8 | 15.2 | 28.7 | 14.6 | 11.0 |
| African dust | 0.61 ± 0.01 | 0.38 ± 0.01 | 0.41 ± 0.02 | 0.19 ± 0.02 | 1.49 ± 0.13 |
| Sea salt | 1.22 ± 0.15 | 0.37 ± 0.10 | 0.42 ± 0.02 | 0.11 ± 0.01 | 0.25 ± 0.03 |
| Wildfires | 0.50 ± 0.02 | NE[2] | NE[2] | NE[2] | NE[2] |
| **Total natural** | **2.33 ± 0.18** | **0.75 ± 0.11** | **0.83 ± 0.04** | **0.30 ± 0.03** | **1.74 ± 0.16** |
| | Relative contributions of natural sources (%) | | | | |
| African dust | 2.3 ± 0.1 | 2.4 ± 0.1 | 1.4 ± 0.1 | 1.3 ± 0.2 | 13.7 ± 1.2 |
| Sea salt | 4.6 ± 0.6 | 2.5 ± 0.7 | 1.5 ± 0.1 | 0.7 ± 0.1 | 2.3 ± 0.3 |
| Wildfires | 1.9 ± 0.1 | NE[2] | NE[2] | NE[2] | NE[2] |
| **Total natural** | **8.7 ± 0.8** | **4.9 ± 0.8** | **2.9 ± 0.2** | **2.0 ± 0.3** | **16.0 ± 1.5** |

[1]Contributions may slightly differ from the values reported in Amato et al. (2016) due to different statistics of the respective datasets.

5    [2]NE: Not estimated.

**Table 4.** Deming regression analysis of dust loads predicted by transport models versus the net dust concentration calculated: (i) through regional $PM_{10}$ concentration data for Athens and (ii) by PMF analysis for Florence. The lower and upper bounds at the 95% confidence interval for the calculated slopes and intercepts are presented in parentheses.

| | | Height | Pearson's coefficient[*] | Slope | Intercept |
|---|---|---|---|---|---|
| **ATHENS** | SKIRON model | Surface | 0.83 | 1.2 (0.9;1.5) | -1.1 (-2.0;-0.1) |
| | | 450 m a.s.l. | 0.86 | 1.9 (1.6;2.2) | -1.8 (-2.9;-0.8) |
| | | 600 m a.s.l. | 0.87 | 2.4 (2.1;2.7) | -2.4 (-3.5;-1.2) |
| | | 750 m a.s.l. | 0.87 | 3.0 (2.6;3.3) | -2.5 (-3.9;-1.2) |
| | DREAM8b v2.0 model | Surface | 0.77 | 2.4 (2.0;2.8) | -1.3 (-2.6;0.0) |
| **FLORENCE** | SKIRON model | Surface | 0.64 | 1.9 (1.1;2.7) | 0.0 (-0.3;0.3) |
| | | 590 m a.s.l. | 0.65 | 2.1 (1.3;3.0) | 0.0 (-0.4;0.3) |
| | | 760 m a.s.l. | 0.66 | 2.5 (1.7;3.4) | -0.1 (-0.5;0.3) |
| | | 940 m a.s.l. | 0.67 | 3.1 (2.1;4.0) | -0.1 (-0.6;0.3) |
| | DREAM8b v2.0 model | Surface | 0.61 | 4.9 (3.0;6.8) | -0.6 (-1.7;0.5) |

[*] All correlations were significant at p = 0.05.

**List of figures**

Figure 1: Map of Europe and AIRUSE cities: Porto (Portugal), Barcelona (Spain), Milan and Florence (Italy) and Athens (Greece).

Figure 2: Annual mean (left) and 90.4th percentile (right) of $PM_{10}$ concentrations (PM10_tot) and the respective adjusted concentrations after subtracting the contribution of natural sources (PM10_adj), for all AIRUSE sites. Red lines denote the annual (40 μg m$^{-3}$) and 24 h limit value (50 μg m$^{-3}$) set by the EU (Directive 2008/50/EC).

Figure 3: Annual mean of $PM_{2.5}$ concentrations (PM2.5_tot) and the respective adjusted concentrations after subtracting the contribution of natural sources (PM2.5_adj), for the AIRUSE sites. Red lines denote the annual limit value (25 μg m$^{-3}$) for $PM_{2.5}$ set by the EU (Directive 2008/50/EC).

Figure 4: Mean source contributions (%) to $PM_{10}$ concentrations during all days (left) and days with exceedance of the 24 h limit value (right). In the case of Barcelona, no exceedance was observed, so days with concentrations greater than the 90th percentile were selected as representative of high pollution days.

Figure 5: Net dust concentrations calculated from regional $PM_{10}$ concentration data: The net dust concentrations when scenarios N2, N3 or N4 are followed are shown in blue. The increment in net dust concentration when a less strict criterion is selected, thus scenarios N1, N2 or N3 are followed, is shown in red.

Figure 6: FLORENCE: Deming regression analysis of Net dust concentrations calculated by the PMF Saharan dust source versus Net dust calculated by the EC methodology with input data: $PM_{10}$ regional background concentrations (left) or the $PM_{10}$ mineral component from stoichiometry (right). The black line corresponds to the linear regression equation, while the red dotted lines are the upper and lower bounds, at the 95% confidence interval.

Figure 7: ATHENS: Deming regression analysis of Net dust concentrations calculated from $PM_{10}$ regional background concentrations ($PM_{10}$) versus Net dust calculated from (i) the PMF mineral dust contributions to $PM_{10}$ (MIN-PMF) (left) and (ii) the $PM_{10}$ mineral component (MIN-STOICH) (right). The black line corresponds to the linear regression equation, while the red dotted lines are the upper and lower bounds, at the 95% confidence interval.

Figure 8: Deming regression analysis of Net dust concentrations calculated from regional background $PM_{10}$ and PMcoarse ($PM_{2.5-10}$) concentrations for Athens (left) and Barcelona (right). The black line corresponds to the linear regression equation, while the red dotted lines are the upper and lower bounds, at the 95% confidence interval.

Figure 9: Deming regression analysis of Net dust concentrations calculated from regional background $PM_{10}$ and PMcoarse ($PM_{2.5-10}$) concentrations for Porto (left) and Milan (right). The black line corresponds to the linear regression equation, while the red dotted lines are the upper and lower bounds, at the 95% confidence interval.

Figure 10: ATHENS: Net dust versus $PM_{2.5}$/$PM_{10}$ concentration ratios, when dust calculation is based on: (i) $PM_{10}$ concentrations (left) and (ii) the $PM_{10}$ mineral component (right).

Figure 11: Source chemical profiles obtained by the application of the PMF model (Amato et al., 2016): Two mineral dust sources were identified in Florence (African dust_FI and Local dust_FI) while a mixed mineral dust profile was found in Athens (Mineral dust_ATH).

Figure 12: Deming regression analysis between net dust calculated through $PM_{10}$ regional background data and dust concentrations modelled at surface level by (a) the SKIRON/Dust and (b) the BSC DREAM8b v2.0 model, for the city of Athens. The black line corresponds to the linear regression equation, while the red dotted lines are the upper and lower bounds, at the 95% confidence interval.

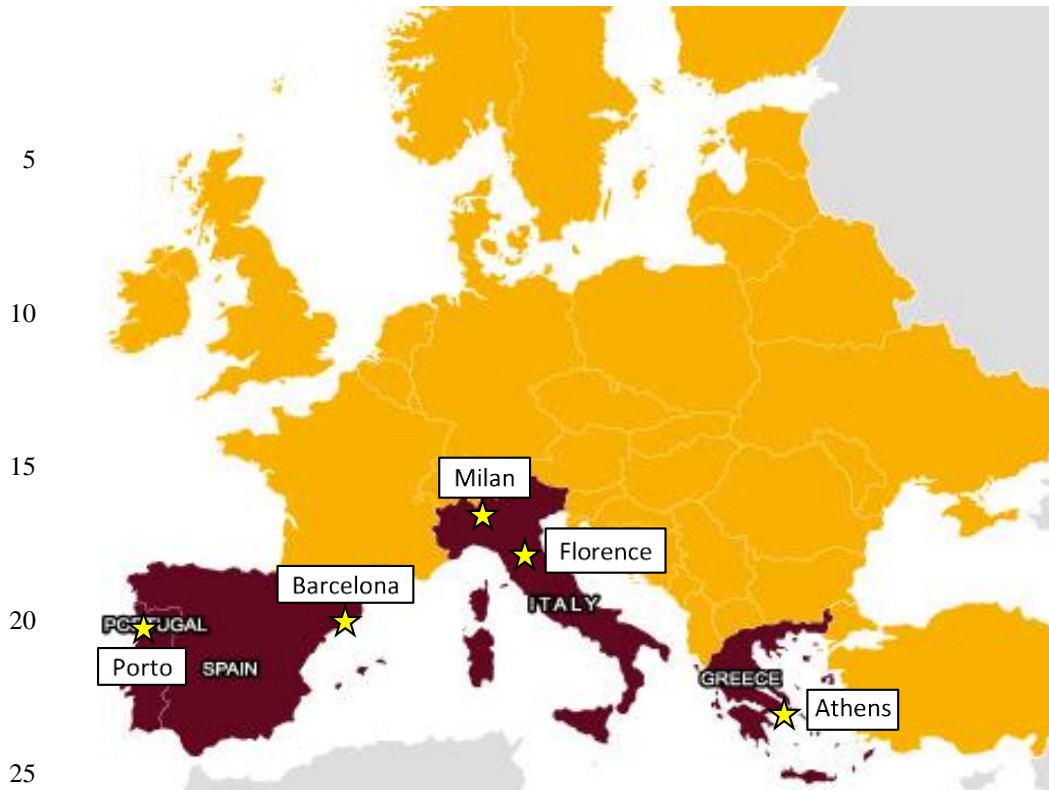

Fig. 1

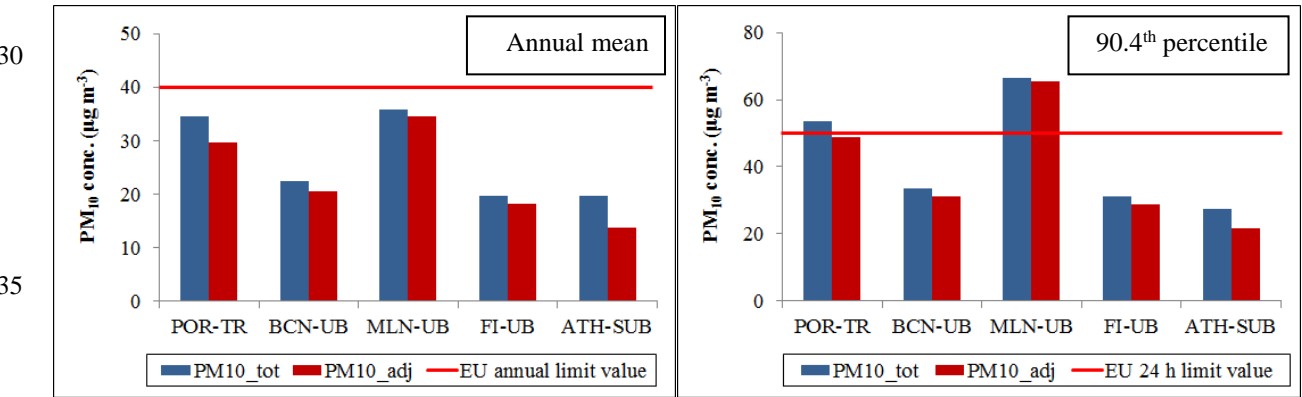

Fig. 2

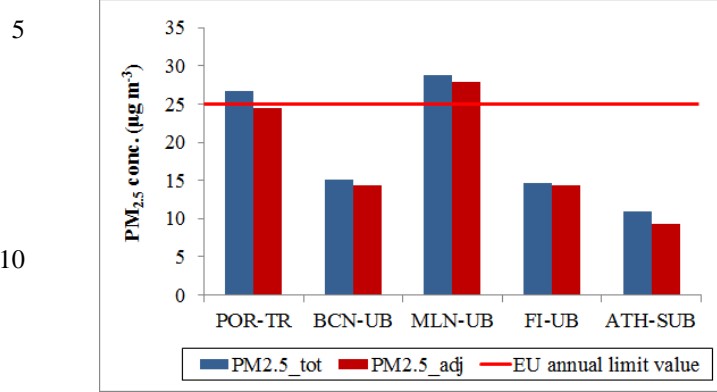

Fig. 3

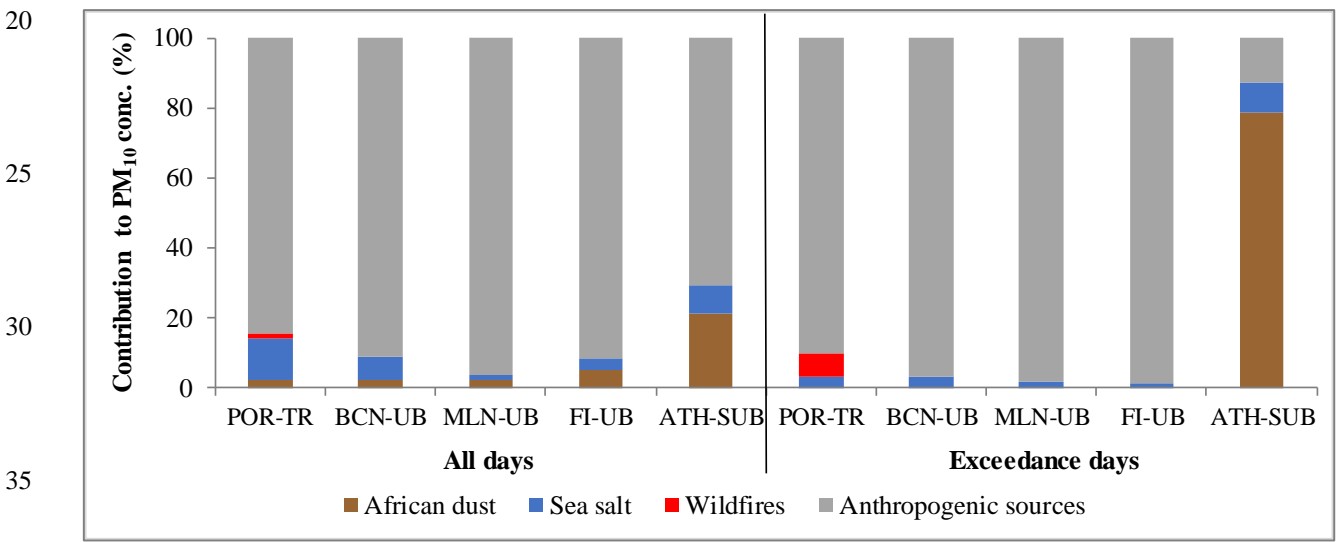

Fig. 4

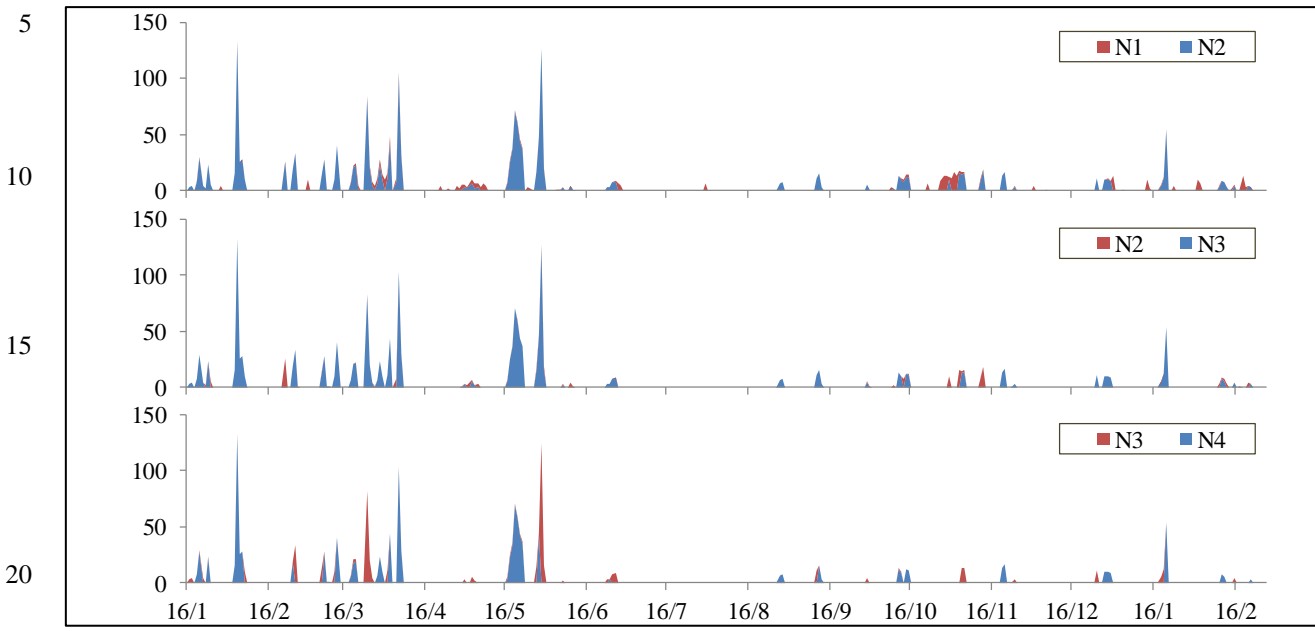

Fig. 5

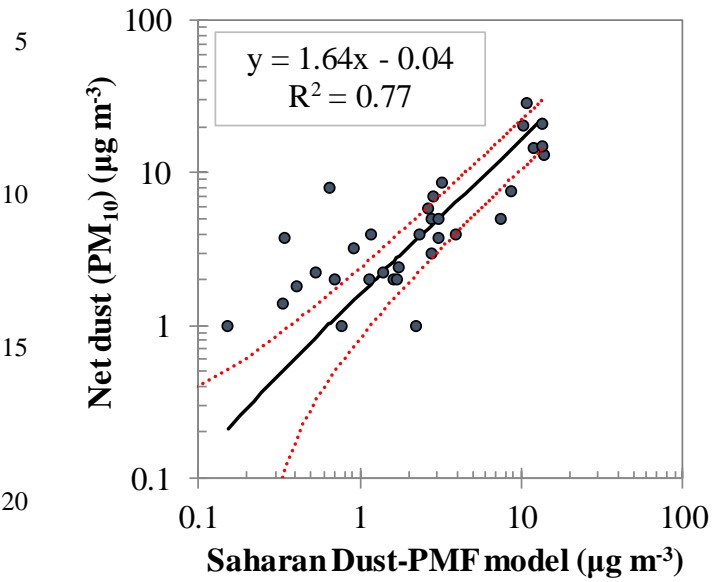

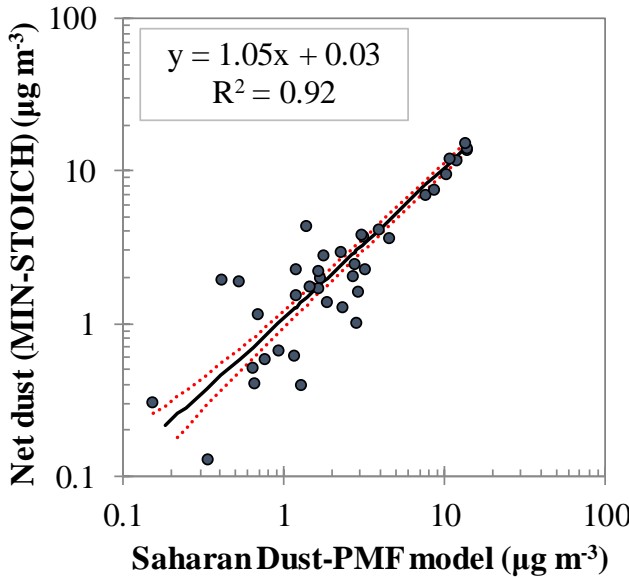

Fig. 6

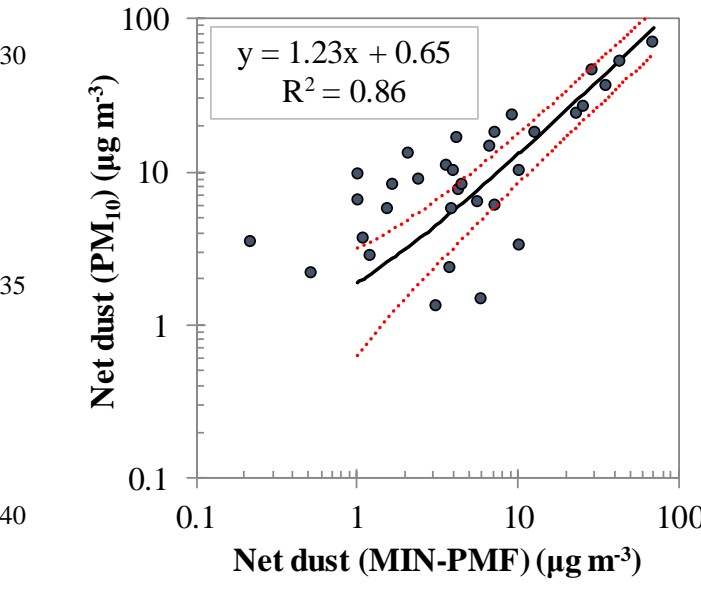

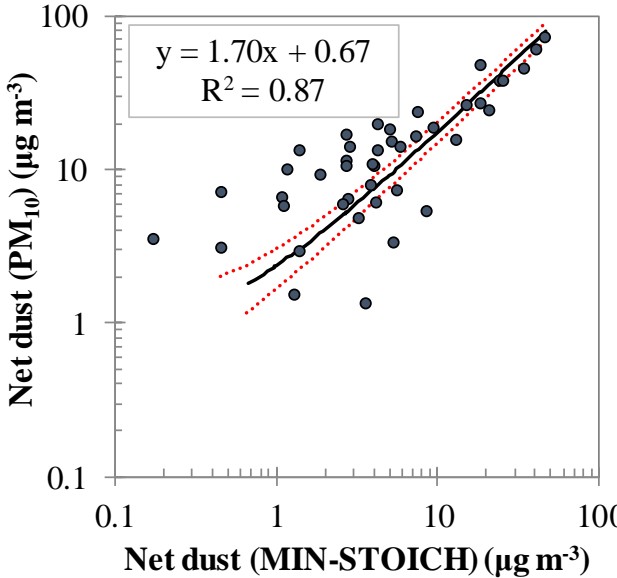

Fig. 7

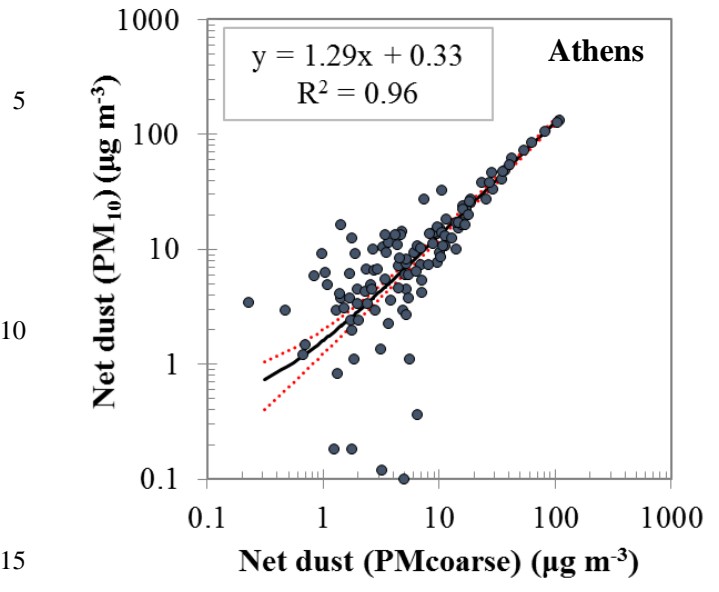
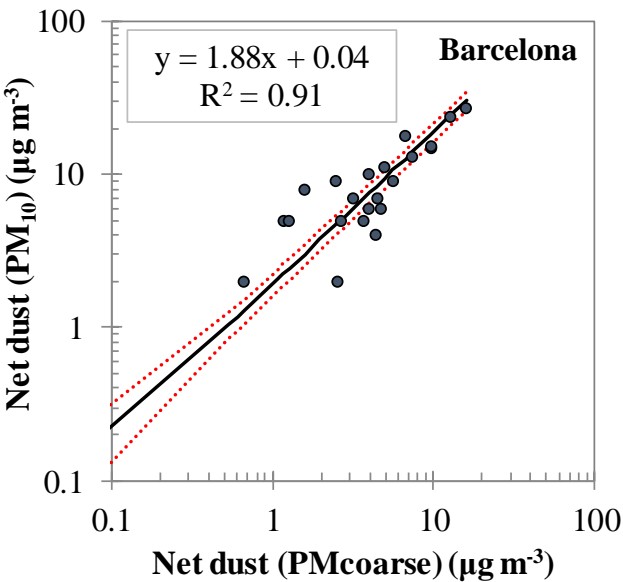

Fig. 8

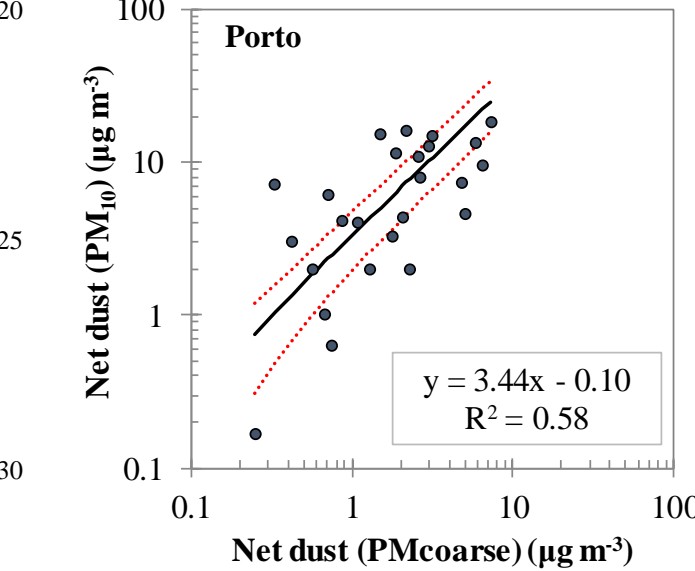
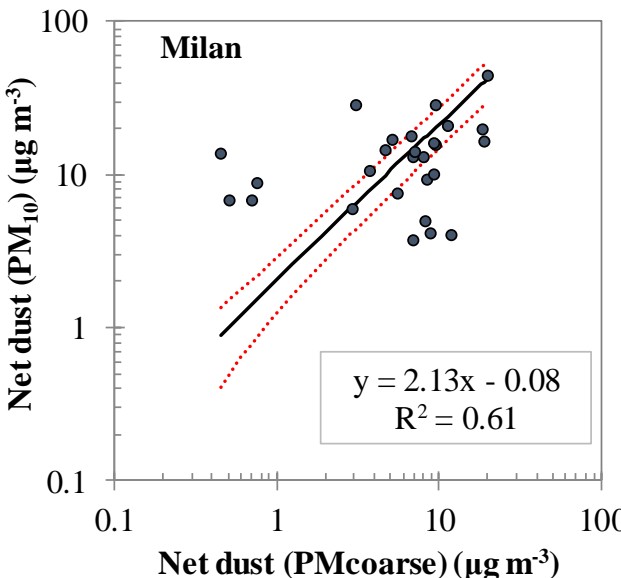

Fig. 9

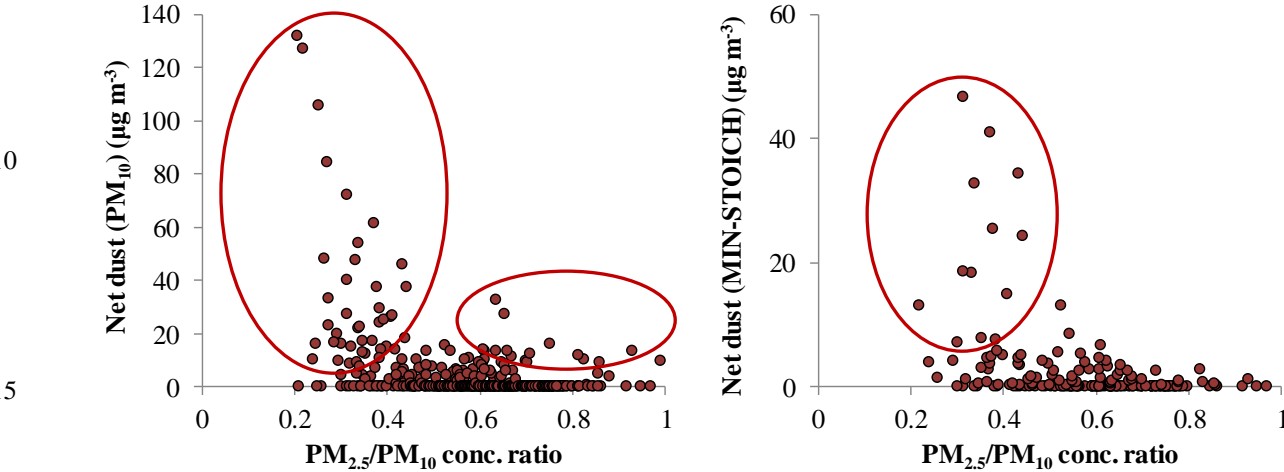

Fig. 10

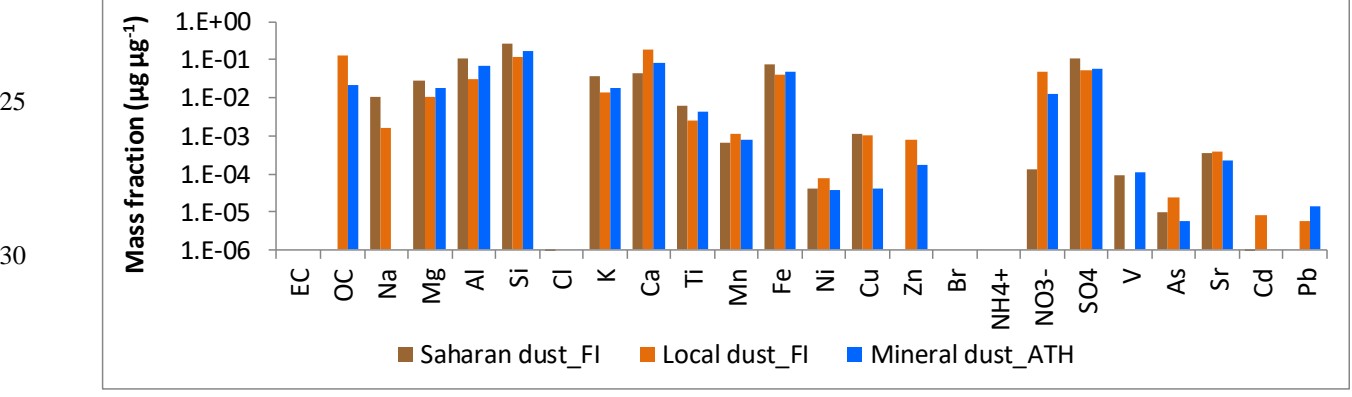

35

Fig. 11

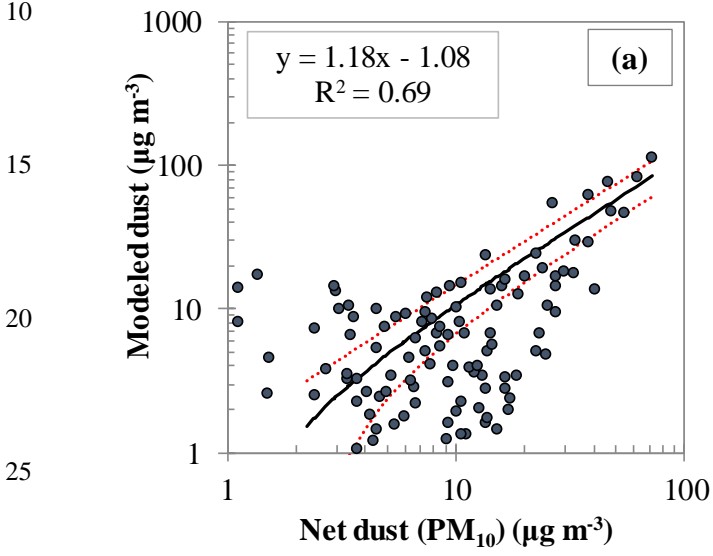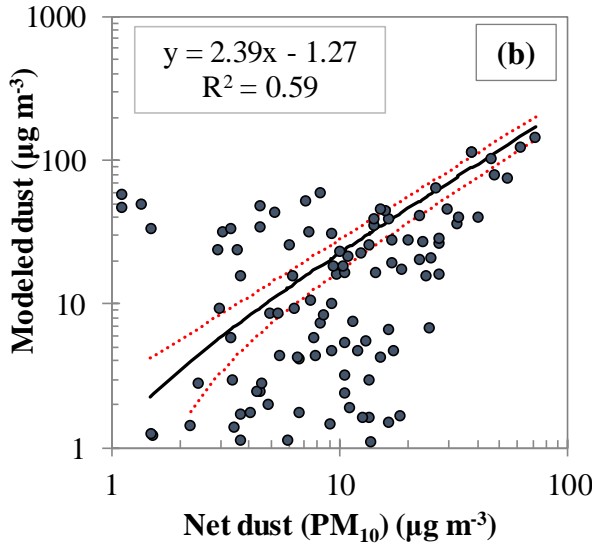

30   Fig. 12