# Peer review of "AIRUSE-LIFE +: Estimation of natural source contributions to urban ambient air $PM_{10}$ and $PM_{2.5}$ concentrations in Southern Europe. Implications to compliance with limit values."

_Atmospheric Chemistry and Physics, 2016_

## Referee Comment (RC1) · Anonymous Referee #2 · 12 Oct 2016

This manuscript describes a well performed analysis of the contribution of natural sources to PM concentrations in 5 cities in Southern Europe. One year of data collection of PM and chemical analysis was performed. The data analysis is based on various state-of-the-art techniques. The results are presented and discussed appropriately. The text is well written and the reasoning well documented. The manuscript is a nice piece of craftmanship. It lacks scientific inspiration or innovation. Considering the Aims and Scope of ACP, i.e. "...The journal scope is focused on studies with general implications for atmospheric science rather than investigations that are primarily of local or technical interest..." (http://www.atmospheric-chemistry-andphysics.net/about/aims_and_scope.html), the manuscript is barely publishable there.

Specific comments:

The reasoning for the obstrusive mentioning of "AIRUSE-LIFE +", even in the title of the manuscript, becomes slightly clear after reading the acknowledgements, but not earlier than that. The acronym should be deleted from the entire manuscript except the acknowledgements.

page 2, line 10: it is not clear what is meant by "This natural background ..." because in the line before, it is talked about "both to natural sources and anthropogenic long range transported particles".

page 4, line 26: how is nss-Na computed?

Fig. 12: Figures should not have headings, just subscript captions

---

## Referee Comment (RC2) · Anonymous Referee #1 · 18 Oct 2016

The paper addresses the question of the natural contributions to PM levels which – although not dealing with novel concepts – has important implications for policy abatement strategies and measures. The paper novelty stands in the attempt of evidencing differences when comparing different approaches and assessing major causes of uncertainties. The paper is clear and well written. The datasets presented are suitable for such kind of analysis. As for the methods used, they are generally scientifically sound although a major concern is related to the algorithm reported for the stoichiometrically derived mineral dust which is not compliant to the mentioned reference and – in general – does not consider Ca, Fe, and K contributions. Maybe that it is simply a typo

error but – if it is not the case – a large part of data analysis should be done again and the text modified accordingly. Another issue concerns the linear regression analyses which should be represented in more suitable way and the equations must be reported with all relevant parameters (e.g. with intercepts, uncertainties and confidence levels).

The referee suggests to accept the paper with major revisions, which should take carefully into consideration the specific comments reported below.

Specific comments: - Please correct the misuse of the possessive case throughout the text (e.g. line 17 page 1 "sources' contribution", line 13 page 3 "pollutants' removal", etc.). - Lines 16-17: Please specify if referring to aerodynamic diameter or other equivalent diameters. - It would be useful for the reader to add references for BSC-DREAM8b and FLEXTRA model. - Page 5 line 13: The algorithm reported in Marcazzan et al. (2001) is not the one written here. Please check it carefully in the original paper by Marcazzan et al. (2001) and change the data/comments accordingly if obtained with the wrong formula. - Line 18 page 5: Here mean contributions for African dust stands for the average obtained considering all the approaches reported in par. 2.2? Please specify. - Line 2 page 6: Please give an explanation for the African dust events during winter in Porto while in Barcelona they were recorded mostly during summer and at the other two cities in springtime. - Figure 4: are you sure that the suburban character of the monitoring site in Athens does not affect the results? The large difference in the proportion between anthropogenic and natural sources is suspicious. - Fig. (not Fog.) 6-9: it is not clear to the referee why the authors represented all these regression lines in a log-log scale. Moreover, 1) the regression lines often show a clear intercept which has not been reported in the regression equation; 2) the values reported for squared-R seem not to correctly represent real data dispersion. How large is the associated uncertainty? How much is this linear regression compatible with a true-linear model? The referee suggests to represent the data in a linear scale, possibly making an orthogonal/Deming regression in order to take into account uncertainties in both x- and y-data as well as the compatibility with a linear model within a given confidence level. Last but not least, check if the MIN-STOICH data reported here have been calculated with the formula reported in the text or using the original Marcazzan et al. algorithm. - Line 28 page 8: also this "dirty" profile for African dust in Athens suggests that the suburban character of the monitoring site may affect the results. Please add a comment in the text. - Table 4: is there any explanation for the relatively higher intercept and slope given by BSC_DREAM model at surface level when compared to SKIRON model? - Figure 12: same comment reported above for Figs. 6-9

---

## Referee Comment (RC3) · Anonymous Referee #3 · 20 Oct 2016

General Comments

This paper deals with desert dust outbreaks in southern Europe, more specifically with the contribution of natural aerosols to mass concentrations measured in five urban environments in Southern Europe. This is an interesting work, well written and very well conducted, with results properly presented and examined (with the exception of the uncertainties on measured and calculated values). In this respect, I really appreciated the sensitivity analysis on the estimation of African dust contributions. However, if this study addresses some relevant scientific questions, many aspects of

desert dust outbreaks in the Mediterranean environment have been broadly studied in recent years (e.g. Stafoggia et al., Environ. Health Perspect., 124 (4), 413-419, 2016 and references therein or Calastrini et al., Advances in Meteorology (2012), http://dx.doi.org/10.1155/2012/246874 and references therein). Therefore, the novelty of this work is limited anyway and it is difficult for me to assess the real contribution of this study to a better knowledge of the Mediterranean atmospheric environment. As the authors pointed out, in the studied urban areas, the natural contribution to the atmospheric particulate load during days in exceedance is very limited, except in Athens, which is not really new (see for example Grivas et al., STOTEN, 389 (2008) 165-177).

From a general appraisal point of view, I suggest to the authors to strengthen their discussion about uncertainties in the quantification of the natural contributions, to reinforce their conclusions, before considering publication of this work in a high ranked journal as ACP.

Specific Comments

Page 4, lines 9 to 13: Please add references about the BSC-DREAM8b and FLEXTRA models.

Page 5, line 13: Equation (2) is not the correct formula reported in the Marcazzan's study! In Marcazzan et al. (2001), the mineral dust concentration is reconstructed from: Mineral Dust = 1.15(1.89Al + 2.14Si + 1.67Ti + 1.4Ca + 1.2K + 1.36Fe).

Please check your "Min-Stoch" data to verify if they have been obtained with the equation (2) or with the original Marcazzan et al. (2001) formula.

Table 2 (page 16) and Table 3 (page 17): Please report uncertainties regarding mass contributions (g.m-3) and relative contributions (%) of natural sources to PM10 and PM2.5 concentrations for the five studied cities.

Figures 6 to 9 (pages 22 to 24): They are clearly intercepts different from 0 in some reported regression lines, which are not considered in the regression equations. . .Could

the authors examine and discuss the impact of these simplifications on their conclusions ?

Page 9, lines 4 to 6 and Figure 12 (page 25): They are undoubtedly no correlation between measured and calculated dust concentrations for concentrations below 10 g.m-3. I suggest to the authors to clearly indicate that in their discussions on the use of the SKIRON and BSC DREAM8b v2.0 models.

Technical corrections

- Page 4, line 27: please change Al for Al in brackets for the non-sea salt Na calculation.
- Page 22: Fig.6 not Fog.6 - Page 25, Fig.11: please, use a log-scaling for the y axis (Mass Fractions), as in Fig.12, for example.
* * *

---

## Author Response (AR1)

**Response to Referees for manuscript ACP-2016-781**

Dear Editor,

Please accept our revised manuscript entitled: "AIRUSE-LIFE +: Estimation of natural source contributions to urban ambient air PM10 and PM2.5 concentrations in Southern Europe. Implications to compliance with limit values". We would like to thank the referees for their constructive comments and suggestions. Our answers to the referees' questions (*in bold italics*) as well as a detailed description of all changes made to the manuscript are included below. Please find attached also our revised manuscript with all changes marked.

We remain at your disposal.

Sincerely,

E. Diapouli

**Anonymous Referee #2**

*This manuscript describes a well performed analysis of the contribution of natural sources to PM concentrations in 5 cities in Southern Europe. One year of data collection of PM and chemical analysis was performed. The data analysis is based on various state-of-the-art techniques. The results are presented and discussed appropriately. The text is well written and the reasoning well documented. The manuscript is a nice piece of craftmanship. It lacks scientific inspiration or innovation. Considering the Aims and Scope of ACP, i.e. "...The journal scope is focused on studies with general implications for atmospheric science rather than investigations that are primarily of local or technical interest..." ([http://www.atmospheric-chemistry-and-physics.net/about/aims_and_scope.html](http://www.atmospheric-chemistry-and-physics.net/about/aims_and_scope.html)), the manuscript is barely publishable there.*

We are obliged to the reviewer for the positive remarks. Although his suggestion on the "barely publishable" quality of the manuscript is vague and unclear, we make an effort below to respond to all the points raised. It is noted that the notion of net dust as described in the current literature has never been tested, as far as we know, against the physical components of dust as described by the stoichiometric calculated component, the PMF derived net dust component and the estimated dust component by transport models like Skiron. All this is done at the same harmonized dataset and for several cities. This kind of sensitivity analysis to our view is innovative, has never been tried before and is introducing new methodology which can be guidance to other studies, as well as having serious policy making impact in South Europe. The continent of Europe and the effect of the Sahara which is the largest dessert on earth can hardly be considered of local interest. We believe that these are the kind of elements that are very well within the Aims and Scope of ACP.

*Specific comments:*

*The reasoning for the obtrusive mentioning of "AIRUSE-LIFE +", even in the title of the manuscript, becomes slightly clear after reading the acknowledgements, but not earlier than that. The acronym should be deleted from the entire manuscript except the acknowledgements.*

It is not understood why the reviewer finds the title of the LIFE+ programme funding the major part of this research obtrusive and requests the removal from the title. This is against ethics of the scientific community where, when the scientific work is the result of an international programme like in this case AIRUSE, its title is often used throughout the text, e.g instead of mentioning the campaigns in each city we refer to them as AIRUSE cities, the results obtained as AIRUSE data and so on. In many cases in the literature large experiments or even smaller ones are mentioned even in the title (see ACE experiments, INDOEX, SUB-AERO, PARTEMIS, etc.). We therefore request to pass on this suggestion by the reviewer as it does not have any scientific nature.

*page 2, line 10: it is not clear what is meant by "This natural background ..." because in*

*the line before, it is talked about "both to natural sources and anthropogenic long range transported particles".*

This phrase has been revised in order to be more precise.

*page 4, line 26: how is nss-Na computed?*

nssNa is computed based on a typical earth's crust elemental ratio (with respect to Al), as:

nssNa = 0.348×Al

This formula is mentioned in the text on Page 4, Line 27 (of the initially submitted manuscript), just below the formula describing the calculation of ssNa.

*Fig. 12: Figures should not have headings, just subscript captions*

The headings have been replaced by relevant labels and the caption has been slightly modified in order to reflect the annotations on the two figures.

**Anonymous Referee #1**

*The paper addresses the question of the natural contributions to PM levels which –although not dealing with novel concepts – has important implications for policy abatement strategies and measures. The paper novelty stands in the attempt of evidencing differences when comparing different approaches and assessing major causes of uncertainties. The paper is clear and well written. The datasets presented are suitable for such kind of analysis. As for the methods used, they are generally scientifically sound although a major concern is related to the algorithm reported for the stoichiometrically derived mineral dust which is not compliant to the mentioned reference and – in general– does not consider Ca, Fe, and K contributions. Maybe that it is simply a typo error but – if it is not the case – a large part of data analysis should be done again and the text modified accordingly. Another issue concerns the linear regression analyses which should be represented in more suitable way and the equations must be reported with all relevant parameters (e.g. with intercepts, uncertainties and confidence levels).*

*The referee suggests to accept the paper with major revisions, which should take carefully into consideration the specific comments reported below.*

The authors would like to thank the reviewer for the suggestions and positive remarks which assisted us in improving the manuscript. We address all general comments and suggestions within the answers given below to the specific comments.

*Specific comments:*

*- Please correct the misuse of the possessive case throughout the text (e.g. line 17 page 1 "sources' contribution", line 13 page 3 "pollutants' removal", etc.).*

The possessive form has been corrected.

*- Lines 16-17: Please specify if referring to aerodynamic diameter or other equivalent diameters.*

At this point in the introduction, the term fine and coarse refers to atmospheric aerosol regardless of equivalent diameter. Equivalent diameters are necessary to consider when we refer to aerosol measured with a specific measurement technique. For example, optical particle sizers also separate fine and coarse particles in terms of their own equivalent optical size. Aerodynamic diameters are relevant to this work because, as can be seen further down, data were obtained by samplers using $PM_{10}$ and $PM_{2.5}$ heads which fractionate particles in terms of aerodynamic diameter, but at this point it is not appropriate to specify this yet. On the other hand, it is trivial to mention that $PM_{10}$ is an aerosol metric by definition referring to aerodynamic diameter.

*- It would be useful for the reader to add references for BSC-DREAM8b and FLEXTRA model.*

References have been added for both models.

*- Page 5 line 13: The algorithm reported in Marcazzan et al. (2001) is not the one written here. Please check it carefully in the original paper by Marcazzan et al. (2001) and change the data/comments accordingly if obtained with the wrong formula.*

The formula proposed by Marcazzan et al. (2001) is:

$$[Mineral\ dust] = 1.15 \times (1.89 \times Al + 2.14 \times Si + 1.67 \times Ti + 1.4 \times Ca + 1.2 \times K + 1.36 \times Fe)$$

Marcazzan et al. (2001) also clarify that only the part of K and Fe of natural origin is included in this calculation. Taking this into account, and considering that Ca, K and Fe have shown to have in the study areas some anthropogenic sources (industrial, construction fugitive sources, traffic and biomass burning), these three elements were replaced in the calculation formula through their typical crustal ratios with respect to Al. For that reason, in the formula we used, Al is multiplied by 3.79 instead of 1.89 (as in the formula proposed by Marcazzan et al., 2001). This methodology has been initially proposed by Nava et al. (2012) and was also adopted in Amato et al. (2016). In the revised text this is better explained and two more references (Nava et al., 2012 and Mason, 1966) were added to Marcazzan et al. (2001), thus clarifying the calculation algorithm used in the present work.

*- Line 18 page 5: Here mean contributions for African dust stands for the average obtained considering all the approaches reported in par. 2.2? Please specify.*

The mean annual contributions of the studied natural sources to $PM_{10}$ and $PM_{2.5}$ concentrations are reported in Tables 2 and 3. This is now clearly stated in the text (in the beginning of section 3.1). In addition, the methodology applied for estimating these contributions is now described in this section.

*- Line 2 page 6: Please give an explanation for the African dust events during winter in Porto while in Barcelona they were recorded mostly during summer and at the other two cities in springtime.*

An explanation has been added, along with a new reference, where the annual cycle of African dust transport is discussed (Moulin et al., 1998).

*- Figure 4: are you sure that the suburban character of the monitoring site in Athens does not affect the results? The large difference in the proportion between anthropogenic and natural sources is suspicious.*

The suburban character of the site does influence the results, especially during exceedance days. The site is not close to direct anthropogenic emissions (as noted in Amato et al., cited in section 2.1), thus exceedances of EU limit values are rare and are almost entirely attributed to African dust events. During the studied year, 79% of the mass concentration during exceedance days was related to African dust

(Page: 7, Lines: 5-6 of the initially submitted manuscript). The suburban character of the site is also commented on the text: "The Athens suburban site on the other hand is a characteristic example of the effect of natural sources in background urban environments." (Page: 7, Lines 3-4 of the initially submitted manuscript). The low levels of $PM_{10}$ at this suburban site definitely govern this behaviour and the numerical results presented here; Sahara dust events are characterized by high $PM_{10}$ concentration values definitely much higher thav the $PM_{10}$ levels at the Athens suburban site and there is therefore nothing suspicious about the fact that almost all exceedances are occurring during these events.

As concentration values, Sahara dust on average provides 4 out of the 20 $\mu g\ m^{-3}$ of $PM_{10}$, as shown in Table 2. However, during exceedance days, the average $PM_{10}$ concentration is 67 $\mu g\ m^{-3}$ out of which 53 $\mu g\ m^{-3}$ is African dust. It has to be noted that dust outbreaks lead to exceedances only in the case of the suburban Athens site.

The significant impact of African dust on $PM_{10}$ concentration levels observed in the city of Athens have been also documented elsewhere. Mitsakou et al. (2008) report on the effects of dust transport on air quality in several Greek urban areas during the period 2003-2006, based on $PM_{10}$ concentration data obtained from stationary monitoring stations and dust concentration data estimated by the SKIRON model. The results show that the monthly mean $PM_{10}$ concentrations measured at a suburban station in Athens have maximum during the month of April, when African dust concentrations are also high. Long-range transport of dust affect the exceedances of the 24 h $PM_{10}$ limit value by 25 and 34% during the spring and autumn periods respectively. In addition, for the year 2003, 65.7% of the daily exceedances are attributed to "African origin".

Mitsakou, C., Kallos, G., Papantoniou, N., Spyrou, C., Solomos, S., Astitha, M., and Housiadas, C.: Saharan dust levels in Greece and received inhalation doses, Atmos. Chem. Phys., 8, 7181-7192, doi:10.5194/acp-8-7181-2008, 2008.

*- Fig. (not Fog.) 6-9: it is not clear to the referee why the authors represented all these regression lines in a log-log scale. Moreover, 1) the regression lines often show a clear intercept which has not been reported in the regression equation; 2) the values reported for squared-R seem not to correctly represent real data dispersion. How large is the associated uncertainty? How much is this linear regression compatible with a true-linear model? The referee suggests to represent the data in a linear scale, possibly making an orthogonal/Deming regression in order to take into account uncertainties in both x- and y-data as well as the compatibility with a linear model within a given confidence level. Last but not least, check if the MIN-STOICH data reported here have been calculated with the formula reported in the text or using the original Marcazzan et al. algorithm.*

The typo has been corrected in Fig. 6. The MIN-STOICH data have been calculated according to Nava et al. (2012), as explained in details above.

All intercepts in the regressions presented in Fig. 6-9 were very low (below 10% of average estimated dust concentrations). Based on the reviewer's suggestions, we have re-analysed the data by applying the

Deming regression and we have included in the revised manuscript all new plots. The correlation coefficients remain the same. Some slopes have changed, while the intercepts are again very low (in some cases lower that the ones calculated through simple linear regression). In the new plots: 1) intercepts are reported along with slopes; 2) Squared-R is also reported; 3) the 95% confidence interval of the regression line is provided.

The log-log scale has been selected for all regressions included in Fig. 6- 9, because of the wide range of values and the high number of zero values (due to the episodic character of African dust events). The reader can have a better visual understanding of the level of discrepancy in the lower values of calculated net dust metrics investigated here and the estimated dust calculated by transport models. This allows the reader to have an understanding of the dust mass concentration levels that this sensitivity analysis is meaningful (mostly $> 5$ µg m$^{-3}$)

We suggest to compare both graphs representations given here and possibly agree with us that the log graph provides a better representation of the relationship between the two parameters compared, especially in displaying the level uncertainty in the lower end of concentration values. In both cases, the Deming regression analysis has been applied, while intercepts are also included.

[Figure]

Figure: Regression analysis of Net dust concentrations calculated from regional background PM$_{10}$ and PMcoarse (PM$_{2.5-10}$) concentrations for Athens, in log-log (left) and linear scale (right). The black line corresponds to the linear regression equation, while the red dotted lines are the upper and lower bounds, at 95% confidence interval.

The advantage of log plots illuminating the concentration levels where the uncertainty on the dust component estimates by the different methods becomes significant, is also identified by reviewer # 3.

*- Line 28 page 8: also this "dirty" profile for African dust in Athens suggests that the suburban character of the monitoring site may affect the results. Please add a comment in the text.*

The chemical profiles depicted in Fig 11 are: 1) the local dust profile for Florence, 2) the African dust profile for Florence and 3) the mixed (local and African) dust profile for Athens (denoted as "mineral dust"). There is no African dust profile for Athens, since we could not separate the local and African dust by PMF analysis in Athens (and similarly in Barcelona, Porto and Milan). This is clearly stated in Page 7, Lines 28-30 and Page 8, Lines 5-6 of the initially submitted manuscript. The Athens mineral dust profile is indeed "dirty", as is the Florence local dust profile. This enrichment with anthropogenic components is already discussed and is found in the mineral dust profiles obtained by PMF for all 5 cities (Amato et al., 2016).

So this is a common finding for all sites and although the urban character of the sites introduces a certain degree of contamination, it is not specific for Athens or the nature of the Athens site. The Saharan dust may be also enriched with anthropogenic components, as shown for the Florence Saharan dust chemical profile (depicted in Figure 11) and documented elsewhere as well (Levin et al., 1996; Sun et al., 2005).

Levin Z., Ganor E. and Gladstein V., (1996) "The effects of Desert Particles Coated with Sulfate on Rain Formation in the Eastern Mediterranean", Journal of Applied Meteorology. 35, pp1511-1523.

Sun Y., Zhuang G., Wang Y., Zhao X., Li J., Wang Z., An Z. (2005) "Chemical composition of dust storms in Beijing and implications for the mixing of mineral aerosol with pollution aerosol on the pathway", Journal of Geophysical Research. 110, D24209, doi:10.1029/2005JD006054.

*- Table 4: is there any explanation for the relatively higher intercept and slope given by BSC_DREAM model at surface level when compared to SKIRON model?*

The differences observed in the slopes and intercepts calculated for SKIRON/Dust and BSC-DREAM8b v2.0 models are related to the parametrizations used by each model for simulating the desert dust cycle, and more specifically with respect to the dust uptake scheme and the soil characterization. This explanation has been also added in the revised manuscript.

*- Figure 12: same comment reported above for Figs. 6-9*

Based on the reviewer's suggestions, we have re-analysed the data by applying the Deming regression and we have included the new plots, in log-log scale. The Deming regression has been also applied for the comparison between the calculated net dust loads and the modelled dust concentrations presented in Table 4. All new results are now included in the revised manuscript. A comparison between the linear and log-log scale figures is given below. We believe that the log-log scale provides a better visual representation of the data.

[Figure]

Figure: Regression analysis between net dust calculated through $PM_{10}$ regional background data and dust concentrations modelled at surface level by SKIRON/Dust for the city of Athens, in log-log (left) and linear scale (right). The black line corresponds to the linear regression equation, while the red dotted lines are the upper and lower bounds, at 95% confidence interval.

**Anonymous Referee #3**

*General Comments*

5   *This paper deals with desert dust outbreaks in southern Europe, more specifically with the contribution of natural aerosols to mass concentrations measured in five urban environments in Southern Europe. This is an interesting work, well written and very well conducted, with results properly presented and examined (with the exception of the uncertainties on measured and calculated values). In this respect, I really appreciated the sensitivity analysis on the estimation of African dust contributions. However,*
10   *if this study addresses some relevant scientific questions, many aspects of desert dust outbreaks in the Mediterranean environment have been broadly studied in recent years (e.g. Stafoggia et al., Environ. Health Perspect., 124 (4), 413-419, 2016 and references therein or Calastrini et al., Advances in Meteorology (2012), http://dx.doi.org/10.1155/2012/246874 and references therein). Therefore, the novelty of this work is limited anyway and it is difficult for me to assess the real contribution of this*
15   *study to a better knowledge of the Mediterranean atmospheric environment. As the authors pointed out, in the studied urban areas, the natural contribution to the atmospheric particulate load during days in exceedance is very limited, except in Athens, which is not really new (see for example Grivas et al., STOTEN, 389 (2008) 165-177). From a general appraisal point of view, I suggest to the authors to strengthen their discussion about uncertainties in the quantification of the natural contributions, to*
20   *reinforce their conclusions, before considering publication of this work in a high ranked journal as ACP.*

We would like to thank the reviewer for highlighting the interesting aspects of this work and we will try to respond to the comment on uncertainty estimation in the specific comments below.

We have to point out that despite the numerous studies addressing African dust outbreaks, this study is
25   one of the few that is based on an organized annual campaign simultaneously in 5 urban areas and also performs an innovative sensitivity analysis of the calculated African dust loads.

*Specific Comments*

*Page 4, lines 9 to 13: Please add references about the BSC-DREAM8b and FLEXTRA models.*

30   A reference has been added for each model.

**Page 5, line 13: Equation (2) is not the correct formula reported in the Marcazzan's study! In Marcazzan et al. (2001), the mineral dust concentration is reconstructed from: Mineral Dust = 1.15(1.89Al + 2.14Si + 1.67Ti + 1.4Ca + 1.2K + 1.36Fe). Please check your "Min-Stoch" data to**
35   **verify if they have been obtained with the equation (2) or with the original Marcazzan et al. (2001) formula.**

We agree with the reviewer. A modified formula has been used for the calculation of the mineral dust concentration in the current work. Marcazzan et al. (2001) noted that only the part of K and Fe of natural origin is included in the calculation of mineral dust concentration. Taking this into account, and considering that Ca, K and Fe have shown to have in the study areas some anthropogenic sources (industrial, construction fugitive sources, traffic and biomass burning), these three elements were replaced in the calculation formula through their typical crustal ratios with respect to Al. For that reason, in the formula we used, Al is multiplied by 3.79 instead of 1.89 (as in the formula proposed by Marcazzan et al., 2001). This methodology has been initially proposed by Nava et al. (2012) and was also adopted in Amato et al. (2016). In the revised text this is better explained and two more references (Nava et al., 2012 and Mason, 1966) were added to Marcazzan et al. (2001), thus clarifying the calculation algorithm used in the present work.

**Table 2 (page 16) and Table 3 (page 17):**

**Please report uncertainties regarding mass contributions (g.m-3) and relative contributions (%) of natural sources to PM10 and PM2.5 concentrations for the five studied cities.**

The uncertainties of the contributions of the different natural sources have been calculated and are reported in Tables 2 and 3. The text has been also modified in order to include information on the methodology used for calculating uncertainty and to comment on the estimated relative uncertainties.

**Figures 6 to 9 (pages 22 to 24): They are clearly intercepts different from 0 in some reported regression lines, which are not considered in the regression equations: Could the authors examine and discuss the impact of these simplifications on their conclusions?**

We agree with the reviewer. The intercepts were very low (below 10% of average concentrations); nevertheless, the regression equations should include both slopes and intercepts. All regressions have been now corrected and the new figures include the slope, intercept and $R^2$.

**Page 9, lines 4 to 6 and Figure 12 (page 25): They are undoubtedly no correlation between measured and calculated dust concentrations for concentrations below 10 g.m-3. I suggest to the authors to clearly indicate that in their discussions on the use of the SKIRON and BSC DREAM8b v2.0 models.**

The phrase has been modified according to the reviewer's suggestion and it is now stated that no correlation was observed between net dust loads and modelled dust concentrations for values below 10 $\mu g\ m^{-3}$, as shown in Figures 12a and b.

**Technical corrections**

**- Page 4, line 27: please change Al for Al in brackets for the non-sea salt Na calculation**.

Corrections have been made to equations (1) and (2).

**- Page 22: Fig.6 not Fog.6**

The correction has been made.

**- Page 25, Fig.11: please, use a log-scaling for the y axis (Mass Fractions), as in Fig.12, for example.**

5    A log-scale is already used for the y axis.

[revised manuscript text omitted]

Fig. 1

[Figure]

Fig. 2

[Figure]

Fig. 3

[Figure]

Fig. 4

[Figure]

Fig. 5

[Figure]

Fog

[Figure]

[Figure]

Fig. 6

[Figure]

[Figure]

[Figure]

[Figure]

Fig. 7

[Figure]

[Figure]

[Figure]

15    Fig. 8

[Figure]

[Figure]

[Figure]

Fig. 9

[Figure]

[Figure]

Fig. 10

[Figure]

Fig. 11

[Figure]

Fig. 12

---

## Author Response (AR2)

**Response to the Editor's Comments for manuscript ACP-2016-781**

Dear Editor,

5  Please accept our revised manuscript entitled: "AIRUSE-LIFE +: Estimation of natural source contributions to urban ambient air PM10 and PM2.5 concentrations in Southern Europe. Implications to compliance with limit values". We would like to thank you for your meticulous editing. We have made all necessary corrections, as shown in the list of comments (*in bold italics*) and replies, attached below. Please also find attached our revised manuscript with all changes marked.

We remain at your disposal.

Sincerely,

15  E. Diapouli

*The authors have reasonably addressed the interactive comments and they have modified their manuscript accordingly.*
*However, in contrast to what the authors claim in their reply to the first comment of Anonymous Referee #1, "sources' contribution" was not corrected everywhere. There are still 5 instances where it is used. I suggest replacing "natural sources' contribution(s)" by "the contribution(s) of natural sources". Also, on page 8, line 3, "models' results" should be replaced by "the results from the various models".*

The appropriate corrections have been made:
- In the title of section 2.2
- In Page 7, Line 10, Lines 15-16, Lines 17-18, Line 18 and Line 20
- In Page 8, Line 3.

*Besides, I have several comments myself that need to be taken into account before this manuscript can be published in ACP.*

*When "sea salt" is used as an adjective there should be a hyphen between the two words; thus it should be replaced by "sea-salt" in this case, starting on page 1, lines 22 and lines 29-30.*

The appropriate corrections have been made in:
- Page 1, Line 22, Line 30
- Page 2, Line 19, 20, 22
- Page 4, Line 24
- Page 5, Line 25
- Page 6, Line 1, 25, 30, 32
- Page 10, Line 16

*Page 1, line 25: Replace "in Eastern" by "in the Eastern".*
The correction has been made.

*Page 2, line 9: Replace "long range" by "long-range".*
The correction has been made.

*Page 3, line 10: Replace "2016" by "2017".*
The correction has been made.

*Page 4, line 9: Replace "by Hybrid" by "by the Hybrid".*
The correction has been made.

*Page 4, line 13: Replace "by Flextra" by "by the Flextra".*
The correction has been made.

*Page 4, line 25: Replace "and earth's" by "and the earth's".*
The correction has been made.

*Page 5, line 7: Replace "on PMF" by "on the PMF".*
The correction has been made.

*Page 5, line 20: Replace "by SKIRON/Dust" by "by the SKIRON/Dust".*
The correction has been made.

*Page 5, line 27: Replace "Wildfires contribution" by "The contribution from wildfires".*
The correction has been made.

*Page 6, line 1: Replace "of African" by "of the African".*
The correction has been made.

*Page 6, line 7: Replace "African dust" by "The African dust".*
The correction has been made.

*Page 6, line 7: Replace "in Eastern" by "in the Eastern".*
The correction has been made.

*Page 6, line 9: Replace "on May" by "in May".*
The correction has been made.

*Page 6, line 11: Replace "dust to" by "dust to the".*
The correction has been made.

*Page 6, line 12: Replace "contributions to" by "contributions to the".*
The correction has been made.

*Page 6, line 13: Replace "between net" by "between the net".*
The correction has been made.

*Page 6, line 15: Replace "High" by "A high".*
The correction has been made.

*Page 6, line 16: Replace "in western" by "in the western".*
The correction has been made.

*Page 6, line 17: Replace "African dust" by "The African dust".*
The correction has been made.

*Page 6, line 18: Replace "during summer" by "during the summer".*
The correction has been made.

*Page 6, line 25: Replace "Sea salt" by "The sea-salt".*
The correction has been made.

5 *Page 6, line 28: Replace "at AIRUSE" by "at the AIRUSE".*
The correction has been made.

*Page 6, line 31: Replace "at the Mediterranean" by "in the Mediterranean".*
The correction has been made.
10
*Page 7, line 1: Replace "Large scale" by "Large-scale".*
The correction has been made.

*Page 7, line 2: Replace "PM levels" by "the PM levels".*
15 The correction has been made.

*Page 7, line 3: Replace "contribution to" by "the contribution to".*
The correction has been made.

20 *Page 7, line 10: Replace "from PM10" by "from the PM10".*
The correction has been made.

*Page 7, line 14: Replace "Similar decrease" by "A similar decrease".*
The correction has been made.
25
*Page 7, line 16: Replace "to marginal" by "to a marginal".*
The correction has been made.

*Page 7, line 19: Replace "for AIRUSE" by "for the AIRUSE".*
30 The correction has been made.

*Page 7, line 22: Replace "Average contributions of natural sources to PM10 concentrations at each"*
*by "The average contributions of natural sources to the PM10 concentrations in each".*
The correction has been made.
35
*Page 7, line 23: Replace "Average" by "The average".*
The correction has been made.

*Page 7, line 26: Replace "In Barcelona" by "In the Barcelona".*
40 The correction has been made.

*Page 7, line 27: Replace "no dusts events" by "no dust events".*

The correction has been made.

*Page 7, line 30: Replace "Exceedances" by "The exceedances".*
The correction has been made.

*Page 7, line 31: Replace "Mean annual" by "The mean annual".*
The correction has been made.

*Page 8, line 2: Replace "AIRUSE" by "the AIRUSE".*
The correction has been made.

*Page 8, line 15: Replace "of (N3)" by "of the (N3)".*
The correction has been made.

*Page 9, line 7: Replace "even higher" by "even a higher".*
The phrase "even higher" was replaced by "an even higher".

*Page 9, lines 9-10: It is unclear why no PM2.5 or PMcoarse data were available for the Florence site. PM2.5 and PM10 samples were collected at this site. Some explanation is needed.*
In the case of Florence, no PM2.5 or PMcoarse data were available from the regional background site. The measurements performed in the framework of this work were at an urban background site. This PM concentration data was not used for the quantification of African dust concentration. The methodology for the calculation of net dust is applied in PM concentrations collected at background sites representative of the regional background concentrations. So, PM10 and PM2.5 data available from the National Monitoring Networks were used for this calculation. This is mentioned in Page 4, Lines 13-17.
The commented phrase has been revised as shown below:
"Florence was not included in this analysis because no PM2.5 or PMcoarse data were available from the regional background site of the National Monitoring Network."

*Page 9, line 27: Replace "by SKIRON/Dust" by "by the SKIRON/Dust".*
The correction has been made.

*Page 9, line 27: Replace "and BSC-DREAM8b" by "and the BSC-DREAM8b".*
The correction has been made.

*Page 9, line 28: Replace "model calculated" by "model the calculated".*
The correction has been made.

*Page 9, line 31: Replace "constrains" by "constraints".*
The correction has been made.

*Page 10, line 1: Replace "results with Athens" by "results as Athens".*

The correction has been made.

*Page 10, line 7: Replace "AIRUSE data" by "the AIRUSE data".*
The correction has been made.

*Page 10, line 9: Replace "that natural" by "that the natural".*
The correction has been made.

*Page 10, line 10: Replace "African dust contribution to PM" by "The African dust contribution to the PM".*
The correction has been made.

*Page 10, line 11: Replace "in Eastern Mediterranean" by "in the Eastern Mediterranean".*
The correction has been made.

*Page 10, line 13: Replace "to PM10" by "to the PM10".*
The correction has been made.

*Page 10, line 15: Replace "western" by "the western".*
The correction has been made.

*Page 10, line 16: Replace "Sea salt" by "The sea-salt".*
The correction has been made.

*Page 10, line 18: Replace "AIRUSE sites" by "the AIRUSE sites".*
The correction has been made.

*Page 10, line 21: Replace "to PM" by "to the PM".*
The correction has been made.

*Page 10, line 23: Replace "contribution" by "the contribution".*
The correction has been made.

*Page 10, line 29: Replace "with input" by "with as input".*
The correction has been made.

*Page 10, line 30: Replace "Analysis of" by "The analysis of".*
The correction has been made.

*Page 11, line 2: Replace "long range" by "long-range".*
The correction has been made.

*Page 11, line 12: Replace ''of BSC-DREAM8b'' by ''of the BSC-DREAM8b''.*
The correction has been made.

*Page 18, line 1: Replace ''to PM10'' by ''to the PM10''.*
5 The correction has been made.

*Page 19, line 1: Replace ''to PM2.5'' by ''to the PM2.5''.*
The correction has been made.

10 *Page 20, line 3: Replace ''95%'' by ''the 95%''.*
The correction has been made.

*Page 20, line 3: Replace ''parenthesis'' by ''parenthesis''.*
The word "parenthesis" was replaced by "parentheses".

*Page 21, line 6: The ''-3'' of ''m-3'' should be in superscript.*
The correction has been made.

*Page 21, line 6: Replace ''EU'' by ''the EU''.*
20 The correction has been made.

*Page 21, line 8: Replace ''AIRUSE sites'' by ''the AIRUSE sites''.*
The correction has been made.

25 *Page 21, line 9: Replace ''EU'' by ''the EU''.*
The correction has been made.

*Page 21, line 19: Replace ''95%'' by ''the 95%''.*
The correction has been made.
30
*Page 21, line 23: Replace ''95%'' by ''the 95%''.*
The correction has been made.

*Page 21, line 26: Replace ''95%'' by ''the 95%''.*
35 The correction has been made.

*Page 21, line 29: Replace ''95%'' by ''the 95%''.*
The correction has been made.

40 *Page 22, line 1: Replace ''of PMF model'' by ''of the PMF model''.*
The correction has been made.

***Page 22, line 5: Replace ''SKIRON/Dust'' by ''the SKIRON/Dust''.***
The correction has been made.

***Page 22, line 5: Replace ''BSC-DREAM8b'' by ''the BSC-DREAM8b''.***
5   The correction has been made.

***Page 22, line 6: Replace ''95%'' by ''the 95%''.***
The correction has been made.

[revised manuscript text omitted]

Fig. 1

[Figure]

Fig. 2

[Figure]

Fig. 3

[Figure]

Fig. 4

[Figure]

Fig. 5

[Figure]

[Figure]

Fig. 6

[Figure]

[Figure]

Fig. 7

[Figure]

[Figure]

Fig. 8

[Figure]

[Figure]

Fig. 9

[Figure]

Fig. 10

[Figure]

Fig. 11

[Figure]

[Figure]

30    Fig. 12

---

## Author Response (AR3)

**Response to the Editor's Comments for manuscript ACP-2016-781**

Dear Editor,

Please accept our revised manuscript entitled: "AIRUSE-LIFE +: Estimation of natural source contributions to urban ambient air PM10 and PM2.5 concentrations in Southern Europe. Implications to compliance with limit values". We have made the necessary corrections, as shown below. Please also find attached our revised manuscript with all changes marked.

We remain at your disposal.

Sincerely,

15   E. Diapouli

*The following alterations are still needed:*
*Page 4, line 7, page 5, line 24, and page 7, line 22: Replace "at each city" by "in each city".*

 The corrections have been made.

[revised manuscript text omitted]

Fig. 1

35

[Figure]

40    Fig. 2

[Figure]

Fig. 3

35

[Figure]

40

Fig. 4

[Figure]

Fig. 5

[Figure]

[Figure]

Fig. 6

[Figure]

[Figure]

Fig. 7

[Figure]

[Figure]

Fig. 8

[Figure]

[Figure]

Fig. 9

[Figure]

Fig. 10

[Figure]

Fig. 11

[Figure]

[Figure]

30    Fig. 12